

# Symmetry-resolved dynamical purification
# in synthetic quantum matter

Vittorio Vitale[1,2], Andreas Elben[3,4], Richard Kueng[5], Antoine Neven[6],
Jose Carrasco[6], Barbara Kraus[6], Peter Zoller[3,4], Pasquale Calabrese[1,2],
Benoît Vermersch[3,4,8] and Marcello Dalmonte[1,2,⋆]

**1** The Abdus Salam International Center for Theoretical Physics,
Strada Costiera 11, 34151 Trieste, Italy
**2** SISSA, via Bonomea 265, 34136 Trieste, Italy
**3** Center for Quantum Physics, University of Innsbruck, Innsbruck A-6020, Austria
**4** Institute for Quantum Optics and Quantum Information of the Austrian,
Academy of Sciences, Innsbruck A-6020, Austria
**5** Institute for Integrated Circuits, Johannes Kepler University Linz,
Altenbergerstrasse 69, 4040 Linz, Austria
**6** Institute for Theoretical Physics, University of Innsbruck, A-6020 Innsbruck, Austria
**7** INFN, via Bonomea 265, 34136 Trieste, Italy
**8** Univ. Grenoble Alpes, CNRS, LPMMC, 38000 Grenoble, France

⋆ mdalmont@ictp.it

## Abstract

When a quantum system initialized in a product state is subjected to either coherent
or incoherent dynamics, the entropy of any of its connected partitions generically in-
creases as a function of time, signalling the inevitable spreading of (quantum) inform-
ation throughout the system. Here, we show that, in the presence of continuous sym-
metries and under ubiquitous experimental conditions, symmetry-resolved information
spreading is inhibited due to the competition of coherent and incoherent dynamics: in
given quantum number sectors, entropy *decreases* as a function of time, signalling dy-
namical purification. Such dynamical purification bridges between two distinct short
and intermediate time regimes, characterized by a log-volume and log-area entropy law,
respectively. It is generic to symmetric quantum evolution, and as such occurs for differ-
ent partition geometry and topology, and classes of (local) Liouville dynamics. We then
develop a protocol to measure symmetry-resolved entropies and negativities in synthetic
quantum systems based on the random unitary toolbox, and demonstrate the generality
of dynamical purification using experimental data from trapped ion experiments [Bry-
dges *et al.*, Science 364, 260 (2019)]. Our work shows that symmetry plays a key role as
a magnifying glass to characterize many-body dynamics in open quantum systems, and,
in particular, in noisy-intermediate scale quantum devices.



## 1 Introduction

Symmetry and entanglement represent two cornerstones of our present understanding of many-body quantum systems. The former governs, e.g., the nature of phases of matter [1–3], while the latter characterizes different classes of quantum dynamics in and out-of-equilibrium [4–6]. Perhaps surprisingly, the intertwined role of these two pillars - falling under the umbrella of symmetry-resolved quantum information - has been relatively unexplored until comparatively recently [7–12]. Such connections are of immediate experimental interest in the context of quantum simulation and quantum computing. Aiming at the ultimate goal of engineering perfectly isolated quantum systems, experiments in synthetic quantum matter and quantum devices realize system dynamics where coupling to an external bath, whatever weak, is *ubiquitous* - two paradigmatic examples being quantum simulators [13, 14] and noisy intermediate-scale quantum (NISQ) devices [15]. In these settings, the microscopic dynamics is local, and is often captured by a master equation with global Abelian symmetries, related to observables such as magnetization or particle number. Against this background, it is an open

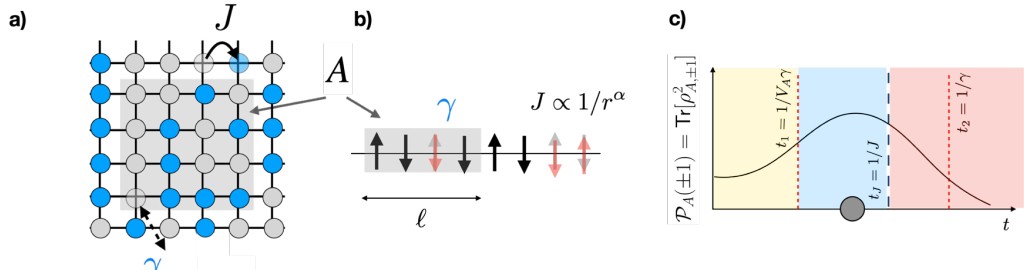

Figure 1: Evolution of symmetry-resolved entropies in NISQ devices. Panel *a,b)*: sketch of the models discussed in the main text. *a)*: free fermions on a square lattice, with tunneling matrix element $J$ and one-body loss rate $\gamma$. *b)*: spin-1/2 chains with long-range XY exchange interactions, and single site spin relaxation rate $\gamma$. The grey areas represent the geometries of the $A$ bipartition of linear length $\ell$ considered below. Panel *c)*: time evolution of the symmetry-resolved purity in the sector $q = \pm 1$, $\mathcal{P}_A(\pm 1) = \text{Tr}[\rho_{A,\pm 1}^2]$, in NISQ devices undergoing three distinct regimes (indicated by different colors, see text for their relations to microscopic timescales). The purity initially increases as a function of time, signalling dynamical purification (gray dot).

question whether symmetry-resolved quantum information can reveal novel, generic classes of quantum dynamics that emerge as a genuine effect of the competition between unitary and incoherent dynamics that is epitomized by quantum simulators and NISQ devices. In fact, symmetry-resolved quantum information has never been studied in the context of open quantum systems.

In this work, we develop a theory and an experimental probe protocol for symmetry-resolved quantum information dynamics in synthetic quantum matter and quantum devices. We are interested in the prototypical scenario depicted in Fig. 1a-b): an initial product state of a lattice model is subjected to the evolution of a $U(1)$ invariant dynamics, where coherent couplings ($J$) are stronger than incoherent ones ($\gamma$). Such scenarios are ubiquitous in current experiment settings, and encompass both interacting and free theories. They are realized in analogue quantum simulators as diverse as trapped ions [16], cold atoms in optical lattices [17], arrays of Rydberg atoms [18], and circuit quantum-electrodynamics settings [19]. Similarly, the interplay between coherent $U(1)$ dynamics and dissipation is of direct relevance to certain nascent quantum computers – those that implement two-qubit SWAP or phase gates with a conserved number of qubit excitations. Concrete examples include architectures based on superconducting qubits [20] and trapped ions [21].

Under these rather ubiquitous conditions, we show that a specific set of symmetry-resolved reduced density matrices undergo *dynamical purification* as a function of time. This phenomenon is strikingly different from purification to an uncorrelated steady state, because it does not come at the expense of quantum information. Using symmetry-resolved negativities, it can be addressed that entanglement remains finite and sizeable over the entire purification dynamics, both in its generic and symmetry-resolved formulation [1]. Furthermore, the scenario we are interested in is fundamentally different from (dissipative) state preparation protocols [23–25] (see below).

The dynamical purification we discuss is at odds with conventional expectations based on information dynamics in many-body systems: starting form low entropy states, Hamiltonian evolution is largely believed to lead to entropy increase, and similar considerations often apply to non-engineered dissipative evolution. What we show is that, for symmetric systems, there

---

[1]We note here that entanglement as witnessed by the negativity requires a different symmetry-resolution with respect to the reduced density matrix [22]. This is due to the presence of partial transposition.

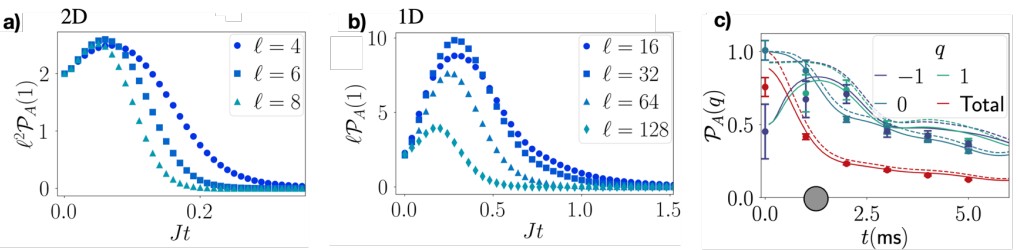

Figure 2: Evolution of symmetry-resolved entropies in NISQ devices. Panel *a,b)*: time evolution of the symmetry-resolved purity normalized by the partition volume, correspondent to a quantum quench from a charge-density-wave state, with the dynamics described in Fig.1a-b), respectively. At short times, decoherences induces a universal scaling behavior, that corresponds to a log-volume entropy scaling, and a purity scaling with inverse of the partition volume. Panel *c)*: symmetry-resolved purity for a long-range XY spin chain of $L = 10$ sites, with $\ell = 4$. The lines represent theoretical simulations, with (solid) and without (dashed) decoherence. Dynamical purification is only present in the first case. Circles represent experimentally reconstructed data from for the symmetry-resolved purity in the trapped ion experiment of Ref. [29]. Dynamical purification is experimentally observed for $q = -1$, and evident for $q = 1$ (albeit with larger error bars) in agreement with both theory and numerics. We refer to Ref. [30] for a thorough statistical analysis.

exist symmetry sectors [2] that evade this scenario. The reason behind this generic - and, we believe, surprising - exception is rooted into the so-far-unexplored combination of the competition between coherent and incoherent dynamics, and the presence of a global conserved charge in a many-body system.

In the exemplifying scenario we anticipated above, we let an initial product state evolve and observe that no purification takes place in the presence of only one of the two contributions (i.e. $J = 0$ or $\gamma = 0$). We note that symmetry plays a crucial role for the effect of such competition to arise, as the latter occurs in reduced density matrices restricted to specific symmetry sectors and is inaccessible in the absence of symmetry resolution. The fact that we need a global charge to be conserved and, most emphatically, that we predict unusual scaling laws for entanglement propagation (see below) allow us to designate dynamical purification as a genuine many-body effect. Thus, dynamical purification is fundamentally distinct from single-body phenomena known in the realm of quantum optics [26–28], such as collapses and revivals, where still purity can increase as a function of time for several reasons. From a more practical viewpoint, dynamical purification can be seen as a direct – and universal – signature of a dominant coherent dynamics in both quantum simulators and NISQ devices, thus providing a simple proxy to evaluate their functioning. Importantly, such phenomenon appears for a broad class of interacting theories, including Hubbard-like, long-range and even confining interactions.

The competition between coherent and incoherent processes reflects into the existence of two distinct dynamical regimes in terms of symmetry-resolved entropy scaling. At short times dissipation is the dominant effect and the symmetry-resolved entropy displays a *log-volume* behavior as function of the volume of the partition where it is computed. At intermediate time, it exhibits a *log-area* one, since coherent dynamics partly overcomes dissipation and enhances purity in given quantum number sectors. The corresponding change of dynamical behavior has dramatic consequences on the experimentally relevant symmetry-resolved purity:

---

[2]In fact, for continuous symmetries, a large majority of the symmetry sectors will display dynamical purification, albeit at different timescales.

the latter quantity scales with inverse volume and inverse area (Fig. 2a-b), respectively. Hence it provides an ideal proxy to diagnose scaling regimes during dynamical purification. For longer time scales, thermodynamics comes back into the game and all symmetry-resolved entropies show the standard extensive behavior in subsystem size [12].

The interplay between the two regimes can be be illustrated in context of a simplified Markovian master equation for the symmetry-resolved reduced density matrix: within that framework, the presence of the coherent dynamics interferes with the action of dissipation and thus leads to a transient regime where entropy is soaked out of the symmetry-resolved reduced density matrix itself. We corroborate our theoretical framework with numerical simulations on a variety of experimentally relevant scenarios. In particular, we showcase the generality of dynamical purification by studying both one- and two-dimensional systems (some of them depicted in Fig. 1a-b) with partitions of different topologies, including both fermionic and bosonic degrees of freedom, and using different types of (weakly-entangled) initial states.

In order to connect our results to experiments, we develop a protocol to access symmetry resolved reduced density matrices building on the random measurement toolbox [31–37]. We show how experimentally demonstrated tools allow for accessing symmetry-resolved moments of symmetry-resolved reduced density matrices and symmetry-resolved Rényi entropies by means of post-selecting data. This procedure is very efficient and allows to reach system sizes that are considerably beyond what can be achieved via full-state tomography, when applicable (See Ref. [38] for a recent demonstration). We apply our protocol to the trapped ion experiment reported in Ref. [29], reconstructing both symmetry-resolved entropies and momenta of the symmetry-resolved reduced density matrix. The experiment reveals a sharp dynamical purification (Fig. 2c) which confirms our theoretical findings. This observation demonstrates the general applicability of our theoretical framework, and concretely illustrates the potential of utilizing symmetry as an enhanced probing tool in state-of-the-art settings.

The paper is organized as follows. In Sec. 2, we set notations and review symmetry-resolved entropies and negativities. In Sec. 3, we specify the time evolution we are interested in, and develop a theory for the time evolution of both entropies and negativities in NISQ devices. We illustrate how entropies show distinct scaling behavior at short (log-volume) and intermediate (log-area) times, so that symmetry-resolved purities actually increase as a function of time (dynamical purification). We then argue that, along this purification, entanglement is typically preserved, so that purification does not take place at the expenses of quantum correlations. In Sec. 4, we present numerical results for both spin chains and fermionic systems supporting our theoretical findings. In Sec. 5, we discuss the protocol for the experimental measurement of symmetry-resolved entropies, and present a first application in the context of the trapped ion experiment, that supports the observation of dynamical purification. Finally, we draw our conclusions.

## 2 Symmetry-resolved quantum information

In this section, we review definitions and properties of symmetry-resolved density matrices and partial transposes. Following those, we introduce symmetry-resolved entropies and negativities, in order to set notation, and briefly discuss applications of such concepts in closed quantum systems.

### 2.1 Symmetry-resolved Renyi entropies

We are interested in bipartite systems, with a partition $A \cup B$. In the case of a many-body pure state, the bipartite entanglement between $A$ and $B$ is fully encoded in the reduced density matrix $\rho_A(\rho_B)$ of the given subsystem $A(B)$, and is characterized via $n$-order Rényi entropies,

defined as

$$S_A^{(n)} = \frac{1}{1-n} \log \operatorname{tr}\{\rho_A^n\}. \tag{1}$$

For $n \to 1$, these reduce to the renowned von Neumann entanglement entropy

$$S(\rho_A) \equiv \lim_{n \to 1} S_A^{(n)} = -\operatorname{tr}_A(\rho_A \log \rho_A). \tag{2}$$

The von Neumann entropy of the reduced density operator is a rigorous entanglement measure for pure states, and the corresponding Rényi entropies with $n > 1$ provide rigorous lower bounds. Both Rényi and von Neumann entropies have found widespread applications in the realm of many-body physics, from the characterization of topological matter, to dynamics out of equilibrium, to the understanding of tensor network methods - see, e.g., Ref. [6] for a review.

For a quantum system whose Hamiltonian dynamics preserves an additive conserved charge, it is possible to identify and compute the contributions to the entanglement related to each symmetry sector [7, 8, 10, 11]. Here, we focus on global symmetries.

Let $Q$ denote such a conserved charge ($Q = Q_A \otimes \mathbb{1}_B + \mathbb{1}_A \otimes Q_B$). Then, the reduced density matrix $\rho_A$ is necessarily block diagonal and each block corresponds to an eigenvalue $q$ of $Q_A$. One can thus introduce $\Pi_q$, the projector into the eigenspace related to eigenvalue $q$, and the associated density matrix $\rho_A(q)$

$$\rho_A(q) \equiv \frac{\Pi_q \rho_A \Pi_q}{\operatorname{tr}\{\rho_A \Pi_q\}}, \quad \operatorname{tr}\{\rho_A(q)\} = 1, \tag{3}$$

so that

$$\rho_A = \oplus_q p(q) \rho_A(q), \tag{4}$$

with $p(q) = \operatorname{tr}\{\rho_A \Pi_q\}$ the probability of being in charge sector $q$. We introduce the symmetry-resolved purity

$$\mathcal{P}_A(q) \equiv \operatorname{tr}\{\rho_A(q)^2\}. \tag{5}$$

It quantifies how mixed the state appears in a given symmetry sector. $\mathcal{P}_A(q)$ ranges in $[2^{-\dim(\mathcal{H}_A(q))}, 1]$ where $\dim(\mathcal{H}_A(q))$ is the dimension of the Hilbert space associated to the symmetry sector $q$ of subsystem $A$.

The symmetry-resolved Rényi entropies are a straightforward extension of this concept:

$$S_A^{(n)}(q) \equiv \frac{1}{1-n} \log \operatorname{tr}\{\rho_A(q)^n\}. \tag{6}$$

Computing $\operatorname{tr}\{\rho_A(q)^n\}$ (in cases when a direct application of projectors in not feasible) requires the knowledge of the spectral resolution in $Q_A$ of $\rho_A$. As pointed out in Refs. [10,11], for some of the computations below, it will be more convenient to study the charged moments $Z_n(\alpha)$,

$$Z_n(\alpha) \equiv \operatorname{tr}\{\rho_A^n e^{i\alpha Q_A}\}, \tag{7}$$

since those do not directly require spectral resolution to start with. The charged moments have been calculated in several cases [10–12, 39–52]. Starting from the computation of $Z_n(\alpha)$, it is possible to obtain $\operatorname{tr}\{\rho_A^n \Pi_q\}$ by means of a Fourier transform:

$$\operatorname{tr}\{\rho_A^n \Pi_q\} = \int_{-\pi}^{\pi} \frac{d\alpha}{2\pi} Z_n(\alpha) e^{-i\alpha q}. \tag{8}$$

We will exploit this last route in the fermionic simulations in Sec. 4.

Recent studies have discussed the basic properties of these symmetry-resolved contributions both in- [10, 11, 39–50, 53] and out-of-equilibrium [12, 51], and in presence of disorder [52]. In basically all considered cases, it has been shown that symmetry-resolved Rényi entropies of large subsystems exhibits entanglement equipartition (namely all symmetry-resolved Rényi entropies are equal) for the most relevant and populated symmetry sectors. The non equilibrium dynamics of symmetry-resolved Rényi entropies has been considered only for isolated systems, both after a local [51] and a global [12] quantum quench, and has revealed the presence of a universal time delay for the activation of a given sector [12]. The investigation of symmetry-resolved Rényi entropies is far from complete and the characterization of its behavior in the presence of dissipation still remains an open question.

## 2.2 Symmetry-resolved entanglement negativity

In the case the system $\mathcal{S}$ is in a mixed state, the entropies of the reduced density matrix are no longer proper measures of bipartite entanglement, as they are also sensitive to classical correlations, although they still provide useful information. A more appropriate and commonly used quantity to witness entanglement in these cases is the negativity [54].

Considering $\mathcal{S} = A \cup B$, according to Peres' criterion [55], also called positive partial transpose (PPT) criterion, a necessary condition for separablity is that the eigenvalues $\lambda_i$ of its partial transpose $\rho^{T_A}$ (with respect to subsystem $A$) are exclusively nonnegative ($\lambda_i \geq 0$). To define $\rho^{T_A}$ we first write $\rho$ as

$$\rho = \sum_{i,j,k,l} \left\langle e_i^A, e_j^B \middle| \rho \middle| e_k^A, e_l^B \right\rangle \middle| e_i^A, e_j^B \right\rangle \left\langle e_k^A, e_l^B \middle| , \tag{9}$$

where $\left| e_i^A \right\rangle, \left| e_j^B \right\rangle$ denote orthonormal bases in the Hilbert spaces $\mathcal{H}_A$ and $\mathcal{H}_B$ corresponding to subsystems $A$ and $B$. Thus one defines the partial transpose $\rho^{T_A}$ performing a standard transposition in $\mathcal{H}_A$, i.e. exchanging the matrix elements in $A$,

$$\begin{aligned} \rho^{T_A} &= (T_A \otimes \mathbb{1}_B)\rho \\ &= \sum_{i,j,k,l} \left\langle e_k^A, e_j^B \middle| \rho \middle| e_i^A, e_l^B \right\rangle \middle| e_i^A, e_j^B \right\rangle \left\langle e_k^A, e_l^B \middle| . \end{aligned} \tag{10}$$

Note that this is equivalent to the basis transformation

$$(\left| e_i^A, e_j^B \right\rangle \left\langle e_k^A, e_l^B \middle|)^{T_A} = \left| e_k^A, e_j^B \right\rangle \left\langle e_i^A, e_l^B \middle| . \tag{11}$$

The entanglement negativity

$$\mathcal{N} \equiv \sum_i \max\{0, -\lambda_i\} = \tfrac{1}{2}\left(\text{tr}\{|\rho^{T_A}|\} - 1\right) , \tag{12}$$

quantifies the degree to which $\rho^{T_A}$ fails to be positive semidefinite. So, a non-zero negativity implies the presence of entanglement between $A$ and $B$. In recent years, the negativity has been extensively studied in a large variety of physical situation, including critical [56–60] and disordered systems [61,62], topological phases [63–67], and out of equilibrium [68–74]. It has been argued that for fermionic systems the partial time-reversal transpose is a more appropriate object to characterise the entanglement in mixed states [75–82], although we will not employ such a concept here.

In analogy to entanglement entropy, one can consider the negativity for a system possessing some additive conserved charge $Q = Q_A \otimes \mathbb{1}_B + \mathbb{1}_A \otimes Q_B$. Interestingly, $\rho^{T_A}$ admits a block

diagonal form in the quantum numbers of the *charge imbalance*, that is, the difference of charge between $A$ and $B$, $\tilde{Q} = Q_A - Q_B^{T_A}$ [22]. Let $\Pi_{\tilde{q}}$ denote the projector onto the eigenspace of $\tilde{Q}$ associated with eigenvalue $\tilde{q}$. We define the normalized symmetry-resolved partially transposed density matrix [22, 83]

$$\rho^{T_A}(\tilde{q}) \equiv \frac{\Pi_{\tilde{q}} \rho^{T_A} \Pi_{\tilde{q}}}{\text{tr}\{\rho^{T_A} \Pi_{\tilde{q}}\}}, \quad \text{tr}\{\rho^{T_A}(\tilde{q})\} = 1, \tag{13}$$

such that

$$\rho^{T_A} = \oplus_{\tilde{q}} \, \tilde{p}(\tilde{q}) \rho^{T_A}(\tilde{q}), \tag{14}$$

with $\tilde{p}(\tilde{q}) = \text{tr}\{\rho^{T_A} \Pi_{\tilde{q}}\} \geq 0$ the probability of being in charge imbalance sector $\tilde{q}$. We can thus define the symmetry-resolved negativity as

$$\mathcal{N}(\tilde{q}) \equiv \frac{\text{tr}\{|\rho^{T_A}(\tilde{q})|\} - 1}{2}, \tag{15}$$

with $\mathcal{N} = \sum_{\tilde{q}} \tilde{p}(\tilde{q}) \mathcal{N}(\tilde{q})$. To compute the symmetry-resolved negativity, one needs the spectral resolution of $\rho^{T_A}$ as in the previous case. Beyond the case of exact simulations, this challenging calculation is performed in two steps. We first focus on the moments $\text{tr}\{(\rho^{T_A}(\tilde{q}))^n\}$, from which the negativity is obtained from a replica trick [57]. Then we consider the charged moments [22, 78]

$$R_n(\alpha) \equiv \text{tr}\{(\rho^{T_A})^n \, e^{i\alpha \tilde{Q}_A}\}, \tag{16}$$

and performing a Fourier transform we get the desired $\text{tr}\{(\rho^{T_A}(\tilde{q}))^n\}$. This way of performing the calculation is very powerful when combined with 1+1D CFTs [22,57], which also provided exact results for the time evolution of the symmetry-resolved negativity after a local quantum quench [51]. As in the case of symmetry-resolved Rényi entropies the study of symmetry-resolved negativities in open systems has never been addressed.

# 3 Time-evolution of symmetry-resolved entropies and negativities

In this section, we present a theoretical description of symmetry-resolved quantum information in NISQ devices. We are specifically interested in the short- to intermediate timescales, that is, before dissipation takes over the system dynamics overwhelming coherent effects.

We shall first discuss the generic setting and subsequently focus on a specific example that presents the generic features we are interested in: the existence of distinct regimes of entropy scaling, dynamical purification, and its interplay with entanglement. While, for the sake of clarity, most of the technical discussion will be based on illustrative examples, we point out that our conclusions are only relying on very generic conditions, that we now specify in the next subsection, 3.1. In the following, we will consider, for the sake of simplicity, $\hbar = 1$ and lattice constant $a = 1$.

## 3.1 Short-time dynamics: emergent purification

The system dynamics we are interested in features the following characteristics:

- a $D$-dimensional system, and a 'convex' partition $A$ herein with smooth boundaries[3], volume $\mathcal{V}_A$ and area $\partial \mathcal{V}_A$;

---

[3]This assumption is only needed to a have simple count of the coherent processes. Essentially, we do not want to have sites of the complement that are accessible from two sites of the partition within first order perturbation theory.

- an initial state $|\psi_0\rangle$ which is a product state in real space;

- the full system dynamics shall be described by a Gorini-Kossakowski-Sudarshan-Lindblad (GKSL) master equation. In particular, we will be interested in Markovian time-evolution;

- the system Hamiltonian shall have a global symmetry $G$. For the sake of simplicity, we will consider $U(1)$ below [4]; most results are immediately extended to $\mathbb{Z}_N$ symmetries, and might also be applicable to the symmetry resolution of continuous non-Abelian groups when sectors are labelled by Abelian subgroups. We assume local (i.e., nearest-neighbor, one- and two-body) couplings, that are homogeneous in space. We define as $J$ the energy scale associated to these terms. Below, we will discuss how sufficiently long-range interactions can also be included;

- dissipation shall instead be described by local (single-site) jump operators. For the sake of simplicity, it is assumed that all sites are affected by the same dissipative processes. Dissipation shall violate the symmetry $G$. We define as $\gamma$ the energy scale associated to these terms, that is, the bare inverse decay rate. One important aspect (that, as we discuss below, is experimentally motivated) is the fact that the jump operators shall still preserve the block structure of the density matrix - a scenario that is typically referred to as weak symmetry [84,85]. Other sources of dissipation can in principle be introduced: as it will be clear below, we expect that their effects are not particularly interesting for the sake of our treatment.

Such assumptions are ubiquitous in the context of synthetic quantum systems, such as cold atoms in optical lattices or tweezers, trapped ions, and arrays of superconducting qubits. Engineering initial states in product state form (up to initialization errors) is of widespread practice, as this can be typically carried out by manipulating quantum states locally. The system dynamics is often local and associated to continuous symmetries, such as particle number or magnetization conservation. Dissipation is generically violating conservation laws associated to the latter quantities: examples include particle loss in cold atom Hubbard models, and fully depolarizing noise and spin relaxation in trapped ions and superconducting circuit architectures.

Most of the present experimental settings are able to access parameter regimes where dissipation is weaker than the coherent dynamics, with the ratio $\gamma/J$ ranging from $10^{-3}$ to $10^{-1}$ [14,17,17]. We will focus explicitly on this parameter regime, and consider dissipation as a perturbation on the top of the coherent dynamics.

Under these assumptions, one can identify three timescales: two intrinsic, and one typical of the subsystem one is interested in. The first one $t_J = 1/J$ is associated to coherent local dynamics. The second one $t_2 = 1/\gamma$ is instead related to a timescale after which (on average) all sites within the partition have undergone a quantum jump. The last one, typical of the subsystem $A$, $t_1 = 1/(\mathcal{V}_A \gamma)$ is related to the timescale required to observe a single quantum jump within $A$.

Let us mention here, that in contrast to the notion of dissipative state preparation [23–25], we study here a given evolution of a physical system. That is, we are not engineering the coupling to the bath to drive the system into a desired state, but rather, discuss the dynamics corresponding to naturally present quantum noise. In addition, whereas dissipative state preparation can be utilized to obtain as a unique stationary state a highly entangled many-body state, or states whose subsystems can be very pure, the situation we consider here is

---

[4] The treatment can be extended to local symmetries, and thus gauge theories. The latter case is more complicated due to the definition of reduced density matrices in Hilbert spaces without tensor-product structure. We leave this to future work, and consider a 1D case in the Sec. 4, that is closer in spirit to the case of global symmetries since Gauss law can be integrated exactly in that case.

not related to long-time dynamics. We will indeed show that dynamical purification occurs at intermediate times.

For times $t \gg t_2$, $\rho_A$ will be completely mixed, also in its symmetry-resolved sectors because the dissipative contribution is overwhelming. Similarly, for regimes where $\gamma \gg J$, the system dynamics is dominated by incoherent processes. A promising regime to observe competition between coherent and incoherent dynamics is thus $\mathcal{V}_A \gamma, J > \gamma$, and is the one we will consider below. We remark that this is a rather generic situation for quantum simulators of many-body systems, where one tries to realize dynamics that are as coherent as possible ($J > \gamma$) for large number of degrees of freedom ($\mathcal{V}_A \gg 1$). This second condition is not needed in general: however, it considerably simplifies the theoretical treatment, as it allows to treat timescales in a way that is easier to interpret. We will thus assume that below, and comment on that at the end of the section. In Sec. 4, we will discuss in more details in which experimental platforms such conditions are met.

We emphasize there that the presence of three dynamical regimes (that, as we will show below, are captured by different entropy scaling) stems from purely geometrical considerations: while Hamiltonian dynamics is acting solely at the boundary between the partitions[5], incoherent processes are instead present over the entire volume of the partition one is interested in. As such, the short-time evolution of symmetry-resolved density matrices will be dictated by this competition, and is expected to be largely insensitive to other characteristics, including the partition geometry and topology, and (to a weaker extent) the initial state. The theoretical apparatus discussed in the next section can be adapted to incorporate such generic features. We nevertheless opted to focus on a simple, yet paradigmatic example, and defer the demonstration of generality of symmetry-resolved dynamical purification to the numerical experiments discussed in Sec. 4.

### 3.1.1 Explicit example: hard-core Bose-Hubbard model in 2D

For the sake of clarity and to make the connections with the numerical experiments below more evident, we start by focusing on a specific instance, and return to the general case at the end of the subsection. We consider a model of hard-core bosons hopping on an infinite 2D square lattice, described by the Hamiltonian

$$H = \frac{J}{2} \sum_{<i,j>} (b_i^\dagger b_j + \text{h.c.}). \tag{17}$$

Here, $b_j$ ($b_j^\dagger$) is the bosonic annihilation (creation) operator at site $j$ such that $n_j = b_j^\dagger b_j$ gives the number operator for that site [6]. The Hamiltonian dynamics conserves the total number of bosons, and is thus $U(1)$ invariant. The system time-evolution is described by a master equation:

$$\partial_t \rho = -i[H, \rho] + \sum_j \gamma \left[ b_j \rho b_j^\dagger + b_j^\dagger \rho b_j - \frac{1}{2} \{ b_j b_j^\dagger + n_j, \rho \} \right], \tag{18}$$

where the second term describes single particle loss and gain processes with decay rate $\gamma$. The full dynamics is schematically depicted in Fig. 3a. While we will keep generality in the theory part with respect to the possible dissipation mechanisms, in the numerical examples below, we will only consider loss terms, as those are more readily accessible experimentally.

---

[5]In the case of sufficiently short ranged power-law interactions, such actions is extended to the few sites close to the boundary.

[6]This choice of jump operators naturally falls within the realm of weak symmetry. In order to break it, one would have to consider jump operators as linear combinations of creation and annihiliation operator, a scenario we do not consider here.

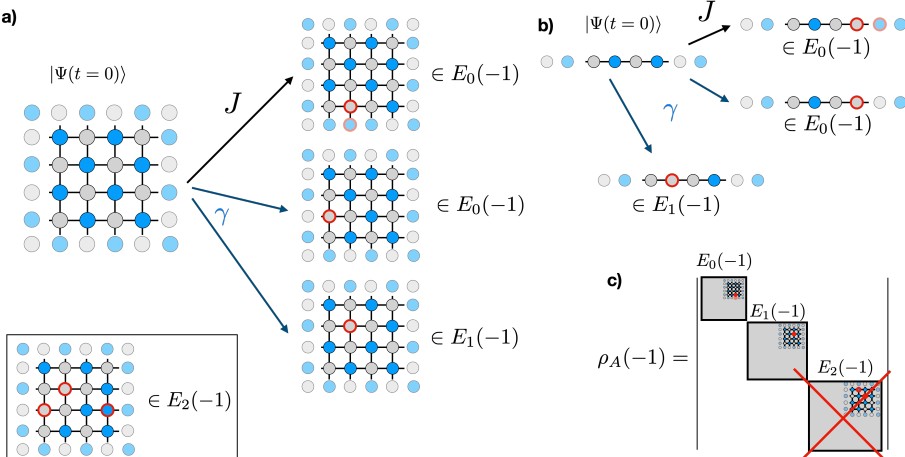

Figure 3: Schematics of the short-time dynamics in lattice models considered here; we show here the sector with $q = -1$. Panel $a$: the system is defined on a square lattice. The initial state is a charge-density wave $|\Psi\rangle$: grey and blue circles represent empty and full sites, respectively. At short time, the evolution involves states belonging to the $E_0$ and $E_1$ subspaces only (see examples, where sites circled in red are the ones changed with respect to $|\Psi\rangle$). The influence of the rest of the Hilbert space $E_2$ on the system dynamics is neglected, as accessing these states will require at least 3 proceeses starting from $|\Psi\rangle$. Panel $b$): same as in panel a, but for the 1D case. Note that, in the following sections, several partition topologies will be discussed. Panel $c$): structure of the time evolved reduced-density matrix. At short times, it is further block-diagonal in both $E_0$ and $E_1$, where the $E_2$ sector is traced away.

We investigate the dynamics starting from a charge-density wave (CDW), with alternating empty (grey) and filled (blue) sites (see Fig.3b). Within this state, we consider the reduced density matrix $\rho_A$ corresponding to a rectangular partition $A$ of size $L_x \times L_y$. Let $Q = \sum_{j \in A} n_j - \frac{1}{2}L_x L_y$ the number of bosons in the partition $A$ above half-filling. Note that, while the full time evolution breaks $U(1)$ invariance, the reduced density matrix $\rho_A$ preserves its block-diagonal form: this is more conveniently seen when interpreting Eq. (18) as a collection of quantum trajectories, each corresponding to the solution of a stochastic Schrödinger equation. Within each trajectory, the total number of particles at each time $t$ is well defined: a single quantum jump only changes that value by an integer value. Following the previous subsection, we denote such symmetry-resolved reduced density matrices as $\rho_A(q)$.

We are interested in short time evolution, where dissipation and coherent dynamics strongly compete. Specifically, we focus on timescales accessible within perturbation theory, that is, $J^2 t^2, t\gamma \ll 1$. Therefore, we can solve Eq. (18) in second order in $t$ to obtain the time-evolved density matrix $\rho(t)$ as a function of the initial state $\rho(0)$ [86]. We focus on the $q = -1$ sector of the reduced density matrix, that is, the one where the number of bosons in the partition is decreased by 1 with respect to half-filling. At short times, this is the most populated sector that does contribute to the initial state. We will comment on the other sectors below. Within this framework, we assume that only the diagonal elements of the reduced density matrix are affected by the time evolution. This assumption can be proven for initial states that are product states in real space.

We now divide $\rho_A(-1)$ into three blocks, schematically depicted in Fig. 3:

1. $E_0(-1)$: states that are connected to the CDW by a single hopping process: these states differ from the CDW by a single occupied site at the boundary. We denote the $(L_x + L_y)$

diagonal eigenvalues of these states as $\lambda_k^{E_0}$;

2. $E_1(-1)$: states that are connected to the CDW by a single pump process in the bulk; these states differ from the CDW by a single empty site in the bulk. We denote the $(L_x - 2)(L_y - 2)/2$ diagonal eigenvalues of these states as $\lambda_k^{E_1}$;

3. $E_2(-1)$: states that are not connected to the CDW by a single tunneling or pump process. We denote the diagonal eigenvalues of these states as $\lambda_k^{E_2}$. For two-body interacting Hamiltonians, these states will be accessed only in third order perturbation theory.

At second order in perturbation theory (the lowest order relevant to the present case), one has the following scaling of the eigenvalues of $\rho_A(-1)$:

$$\lambda_k^{E_0} = (J^2 t^2 + \gamma t)/A(t), \ \lambda_k^{E_1} = \gamma t/A(t), \tag{19}$$

with $\lambda_k^{E_2} = 0$, and

$$A(t) = \gamma t (L_x L_y - 4)/2 + J^2 t^2 (L_x + L_y). \tag{20}$$

We can now compute the time-evolution of the symmetry-resolved entanglement entropies. At short times $t < t_1$, only dissipation is relevant. In particular, the rank of the reduced density matrix will be $(L_x L_y)$, and the corresponding Renyi-2 entropy results:

$$S_A^{(2)}(q = -1) \propto \log[L_x L_y] \qquad \text{for } t \ll t_1, \ L_x, L_y \gg 1, \tag{21}$$

and is time-independent. The corresponding purity is:

$$\mathcal{P}_A(-1) \propto 1/[L_x L_y]. \tag{22}$$

It is worth noting that such 'log-volume' regime is valid at arbitrarily small times, the simple reason being that the initial state has no component in the $q = -1$ subspace.

At intermediate times $t_1 < t < t_J$, tunneling affects the system dynamics. While unitary time evolution generically leads to further information propagation and, correspondingly, entropy production, here, the opposite takes place: the symmetry-resolved density matrix *purifies* as a function of time, i.e., the purity increases and the entropy decreases. The reason for this phenomenon stems from the natural competition between volumetric and perimetral contributions to the system dynamics: while dissipation has an effect that scales with the volume of the partition, and thus populates a number of eigenvalues that are proportional to the volume itself, short-time coherent dynamics is related to boundary effects, and thus favors a much smaller number of states within the Hilbert space of the partition.

In order to elucidate this effect, we observe that our reduced density matrix is already normalized, and compute

$$\mathcal{P}_A(-1) = \frac{t^2}{A(t)^2}[(L_x + L_y)(J^2 t + \gamma)^2 + (L_x - 2)(L_y - 2)\gamma^2],$$

that, in the large volume limit becomes

$$\mathcal{P}_A(-1) \ \simeq \ 1/(L_x + L_y) \ \text{ for } \ t_J > t > t_1.$$

The corresponding Renyi-2 entropy follows a 'log-area' scaling:

$$S_A^{(2)}(q = -1) \propto \log[L_x + L_y] \ \text{ for } \ t_J > t > t_1.$$

This implies that the transition between the two regimes is characterized by an emergent purification, that transits the system from a purity that is inversely proportional to the volume

of the partition, to one that is inversely proportional to its surface. Note that the explicit time evolution can be computed from the previous equation, and in principle, the position of the 'maximum' of the purification can be extracted. While the corresponding formulas reveal no more physical insight, they signal the fact that the purification time decreases as the partition size increases. Due to the condition $t_J < t_2$, it is not possible to analytically compute the $\mathcal{V}_A \to \infty$ limit; we nevertheless expect dynamical purification to systematically decrease with the partition size, as a consequence of the area versus volume competition.

The calculation above can be straightforwardly generalized to any dimension, modulo the conditions mentioned at the beginning of the section, under the assumption that dynamics is acting non-trivially at the boundary (e.g., a state with a layer of empty sites at the boundary will not experience any meaningful coherent evolution at short times in the $q = -1$ sector). The corresponding scaling behavior decomposes into three regimes:

$$\mathcal{P}_A(-1) \propto \begin{cases} 1/\mathcal{V}_A & t_1 \gg t \geq 0 \text{ (short time)}, \\ 1/(\partial \mathcal{V}_A) & t_J > t > t_1 \text{ (int. time)}, \\ 1/2^{\mathcal{V}_A} & t \gg t_J \text{ (long time)}. \end{cases} \tag{23}$$

This equation succinctly describes the dynamical scaling regimes depicted in Fig. 1. Starting from an unsurprising short time behavior (top case), the system purifies at intermediate time scales (center case) before eventually getting fully mixed (bottom case) due to both coherent and incoherent system dynamics.

### 3.1.2 General remarks: nature of interactions, initial state, and dissipation

In the explicit example before, we have focused on the most populated sector of the reduced density matrix not present in the initial state, we expect dynamical purification to occur also in other sectors - with, however, a weaker effect due to higher order perturbative processes. The presence of long-range interactions that decay fast enough (at most as power law) shall not change this picture at the qualitative level: however, it will lead to a renormalization of the timescale $t_J$. Importantly, long-range interactions will not modify the structure of the Hilbert subspaces discussed above.

While we have focused on purities, additional information can in principle be obtained from the population of the different sectors (denoted with $A(t)$ above) as well. One example is equilibration at long-times: this is beyond the perturbative treatment we have developed, and will be discussed in the next sections in both simulations and experiments.

The treatment above is specific to an initial state: however, the competition between volumetric and perimetral contributions is in fact generic to a much broader set of experimentally relevant configurations. For the case of pure states, dynamical purification shall occur as long as the initial state is separable or weakly entangled, as we show in one of the fermionic examples below. For highly entangled initial states, the theory above is not immediately applicable. Below, we will discuss a 1D numerical example, where the initial state has $\log(\ell)$ entanglement: in that case, we observe no purification. It is also important to stress that, while we have assumed $\gamma \mathcal{V}_A > J$, this is technically not needed at all: indeed, since dissipation acts already at first order in perturbation theory, there exists always a time scale for which $\gamma \mathcal{V}_A > J^2 t$. In fact, the size of the partition is irrelevant, as long as it can host significant dynamics within a given symmetry sector.

Most importantly, dynamical purification is present also for initial states that are globally mixed. In those cases, this is simply due to the fact that the coherent dynamics selects a subset of states in $\rho_A(-1)$ that are populated due to coupling to $\bar{A}$. The extent of the dynamical purification depends on the details of the action of the Hamiltonian on the initial state: we will investigate a specific scenario below while discussing trapped ion experiments.

Another aspect that is worth discussing is, which type of noise leads to dynamical purification. The noise we have considered here has two characteristics: (i) it is described by a Markovian master equation, and (2) it is quantum noise, as testified by the fact that it is described by non-hermitian jump operators. While these conditions are typically very well satisfied when describing the dynamics of cold atoms in optical lattices using a master equation, we find useful to provide a short discussion of these two elements in view of possible applications to other settings.

The first assumption above is delicate. Since we are interested in intermediate time dynamics, ht is reasonable to expect that our findings will not be affected by a bath featuring short-lived memory effects, as long as the weak system-bath approximation (that we nevertheless consider, since $\gamma \ll J$) holds. However, more complicated bath structures including strong memory effects - such as a low-temperature Ohmic bath - cannot be immediately connected to the physical picture we present here. We leave this interesting question - that does not pertain the experimental systems we are interested in - to future work.

The second assumption above is crucial: Hermitian jump operators (such as those, for instance, describing classical noise) would not lead to any dynamical purification. This can be easily seen by considering the action of dephasing on the various sectors of the symmetry-resolved reduced density matrix: for the type of initial states we consider, the latter will not affect populations. This implies that entropy will be dominated by coherent dynamics, thus increasing with time. The relevance of the first assumption can potentially be exploited as a diagnostic in the context of quantum noise tomography; interestingly enough, such a probe would be very sensitive, as the effects we describe can be present for very small values of $\gamma$, and can be tuned by changing the volume of the partition in numerical simulations as well as in experiments.

Finally we find it useful to add two comments framing the physics we observe in the context of open quantum systems (especially since the primary physical platform we are interested in are nothing but many-body quantum optical systems, as exemplified by the experimental results below). First, we observe that the effective dynamics describing the evolution of $\rho_A$ can be interpreted as the time evolution of a density matrix of a system coupled to a bath. This provides an additional viewpoint on the phenomenon we are interested in, that could be of help to translate it to other contexts (for instance, in case the two partitions are made of two different types of degrees of freedom, e.g., describing light-matter interactions). A detailed discussion of this fact is provided in the appendix, together with a proof of the fact that such effective dynamics is Markovian. Second, we point out that the phenomenology we describe here is fundamentally distinct from other instances of open system dynamics that may feature a decrease in entropy. One example here are revivals in the Jaynes-Cummings model [27] (and similar effects in the context of non-Markovian dynamics of single spins coupled to cavity modes): there, the phenomenon one observes is intrinsically few body, and has no relation whatsoever with (continuous) symmetries. This fundamental difference is clearly apparent into the fact that the universal regimes we have proposed have not been reported so far in those contexts.

## 3.2 Negativity over dynamical purification

While the system purifies at short time, due to its coupling to the environment, it cannot be established *a priori* whether this is associated to a loss of shared entanglement between the partition and its complement. For instance, dynamical purification (with or without symmetry resolution) can also occur at long times in systems under the presence of dissipation only: a typical example is relaxation to a vacuum state, that is driven by a single jump operator, and leads to a trivial state, with no left-over correlations between $A$ and $B$, and within $A$. Below, we show explicitly how symmetry-resolved dynamical purification is drastically distinct from this

mechanism: In particular we show how not only entanglement between $A$ and $B$ is generated as a function of time, but also that, in any given symmetry sectors (now labeled by quantum number differences), entanglement remains finite and sizeable (negativity of order 1) over the entire purification process. This is a key element that characterizes this symmetry-resolved phenomena, and we will show below how this is also captured within perturbation theory.

We study the entanglement dynamics for two connected partitions $A$ and $B$ of a spin (or hard-core boson) system, as governed by Eq. (18), in a regime where the partition $A$ undergoes dynamical purification. For the sake of simplicity, we will deal explicitly with the 1D case analog to the setup described above (see Fig. 3b), and restrict the decoherence channels to particle loss, as this will allow us to keep our notations compact. Our findings are however general, as illustrated in the next section for various geometries and partition configurations.

In contrast to the situation of dynamical purification, the key features of short-time entanglement dynamics of the partial transpose reduced density matrix can already be captured by solving Eq. (18) in first-order perturbation theory, i.e. by studying the dynamics of $\rho(t)$ in first order in $t \ll 1/(\gamma N), 1/J \ (\ll 1/\gamma)$. We rewrite this as

$$\rho(t) = \rho(0) - i[H, \rho(0)]t + \gamma t \sum_j \left( b_j \rho(0) b_j^\dagger - \tfrac{1}{2} b_j^\dagger b_j \rho_0 - \tfrac{1}{2} \rho_0 b_j^\dagger b_j \right). \qquad (24)$$

Consider for concreteness that the even sites $2m$, $m = 1, \ldots, N/2$ are occupied and, $N_A = N_B$ is even, the density matrix in first-order perturbation theory can be re-expressed as [7]

$$\rho(t) = \left( 1 - \frac{N\gamma t}{2} \right) \rho(0) + Jt(-i b_{N_A+1}^\dagger b_{N_A} \rho(0) + h.c) + \gamma t \sum_{m=1}^{N/2} b_{2m} \rho(0) b_{2m}^\dagger + \ldots, \qquad (25)$$

which corresponds to a diagonal part parametrized by the decoherence rate $\gamma$, and a pair of off-diagonal elements associated with the hopping $J$. Note that there is no diagonal contribution due to the hopping, as this only appears in next-to-leading order as discussed above. Taking the partial transpose of Eq. (25) leads to

$$\rho^{T_A}(t) = \left( 1 - \frac{N\gamma t}{2} \right) \rho(0) + Jt(-i b_{N_A+1}^\dagger \rho(0) b_{N_A}^\dagger + h.c) + \gamma t \sum_{m=1}^{N/2} b_{2m} \rho(0) b_{2m}^\dagger, \qquad (26)$$

which has a 3-block structure associated with the quantum number $\tilde{q} = q_A - q_B$

$$\begin{aligned}
\rho^{T_A}(\tilde{q} = 0, t) &= \left( 1 - \frac{N\gamma t}{2} \right) \rho(0), \\
\rho^{T_A}(\tilde{q} = -1, t) &= \gamma t \sum_{m=1}^{N_A/2} b_{2m} \rho(0) b_{2m}^\dagger + Jt(-i b_{N_A+1}^\dagger \rho(0) b_{N_A}^\dagger + h.c), \\
\rho^{T_A}(\tilde{q} = 1, t) &= \gamma t \sum_{m=N_A/2+1}^{N} b_{2m} \rho(0) b_{2m}^\dagger.
\end{aligned} \qquad (27)$$

The sector $\tilde{q} = 0$ corresponds to the initial state component, has a weight $\text{tr}(\rho^{T_A}_{\tilde{q}=0}(t))$ of order 1, and features no entanglement. The sector $\tilde{q} = -1$, corresponding to the situation where the $A$ partition loses one excitation with respect to partition $B$, has the richest structure, representing the interplay between particle loss from $A$ and coherence dynamics (hopping from $A$ to $B$).

---

[7]For the sake of clarity, we do not include intra-partition hopping terms: at lowest order, their only effect is to renormalize the dynamics in the $\tilde{q} = 0$, that we are not immediately interested here

Finally, the last sector $\tilde{q} = 1$ represent decoherence events occurring in the $B$ partition. In each sector, we can calculate the spectrum

$$\tilde{\lambda}(\tilde{q}=0,t) \;=\; \left(1 - \frac{N\gamma t}{2}\right), \tag{28}$$

$$\tilde{\lambda}^{(m=1,\ldots,\frac{N_A}{2}-1)}(\tilde{q}=-1,t) \;=\; \gamma t, \tag{29}$$

$$\tilde{\lambda}^{(m=\frac{N_A}{2},\frac{N_A}{2}+1)}(\tilde{q}=-1,t) \;=\; (\gamma \pm \sqrt{\gamma^2+4J^2})\frac{t}{2} \approx (\frac{\gamma}{2} \pm J)t, \tag{30}$$

$$\tilde{\lambda}^{(m=1,\ldots,\frac{N_B}{2})}(\tilde{q}=1,t) \;=\; \gamma t, \tag{31}$$

where, in the last part of the third line, we have neglected the term $\gamma^2 \ll J^2$.

The existence of a negative eigenvalue $\tilde{\lambda}^{(m=N_A/2)}(\tilde{q}=-1) = -Jt + \gamma t/2 \approx -Jt < 0$ demonstrates that the state is entangled, and remains so over dynamical purification. After normalization, we obtain the symmetry-resolved negativity

$$\mathcal{N}(\tilde{q}=-1) \approx \frac{2Jt}{N_A \gamma t} = \frac{2J}{N_A \gamma}, \tag{32}$$

that features a characteristic $1/\gamma$ scaling (that is reminiscent of the fact that this is a perturbative effect). Interestingly, the short-time behavior of the negativity is constant: this is consistent with the fact that only one, large negative eigenvalue dominates its behavior. The inverse scaling with the partition size is due to the fact that we are normalizing symmetry-resolved density matrices, so boundary contributions to the entanglement generated by the mechanism described above are expected to be of order $1/N_A$.

## 4 Numerical results

### 4.1 Spin chains

In this section, we provide numerical evidence for symmetry-resolved purification in one-dimensional spin chains. Specifically, we consider quench dynamics in the XY-model with Hamiltonian

$$H_{\text{XY}} = \sum_{i>j} J_{ij}(\sigma_i^+ \sigma_j^- + \sigma_i^- \sigma_j^+) + \sum_i \delta_i \sigma_i^z, \tag{33}$$

where $\sigma_j^{\pm} = (\sigma_j^x \pm i\sigma_j^y)/2$, subject to spin excitation loss with rate $\gamma$, modeled via the jump operators $\sqrt{\gamma}\sigma_i^-$ for $i = 1,\ldots,N$. The coherent hopping is here determined by the interaction matrix $J_{ij}$. We consider next-neighbour interactions, $J_{i,j} = J\delta_{i+1,j}$, and long-range interactions with power law coefficient $\alpha$, $J_{i,j} = J/|i-j|^\alpha$, respectively. A disordered longitudinal field $\delta_i$ can be added which we sample independently for each lattice site from the uniform distribution on $[-\delta,\delta]$. We initialize the system with $N = 8$ sites, divided into subsystems $A = [1,2,3,4]$ and $B = [5,6,7,8]$, in the Néel state $|\Psi_0\rangle = |\downarrow\uparrow\rangle^{\otimes N/2}$ with total magnetization $S_z = \sum_{i=1}^N \sigma_i^z = 0$. While the total magnetization is conserved by the Hamiltonian part of the dynamics $H_{XY}$, the incoherent spin excitation loss leads to a population of various sectors.

In Fig. 4 (a,d), we display the symmetry-resolved purity of the subsystem $A$ with $N_A = 4$ sites for various sectors $q$ in a system with short-range interactions $J_{ij} = J\delta_{i-1,j}$ and vanishing disorder $\delta/J = 0$. Clearly, the sector $q = -1$ exhibits dynamical purification at times $Jt \approx 1$ which is absent in the sector $q = 0$ and also for the purity $\text{tr}[\rho_A^2]$ of the total density matrix $\rho_A$. Note that the second peak in panel (a) (at around $Jt = 4$) is a boundary effect due to the partition size. As predicted by perturbation theory [Eq. (23)], purification is pronounced

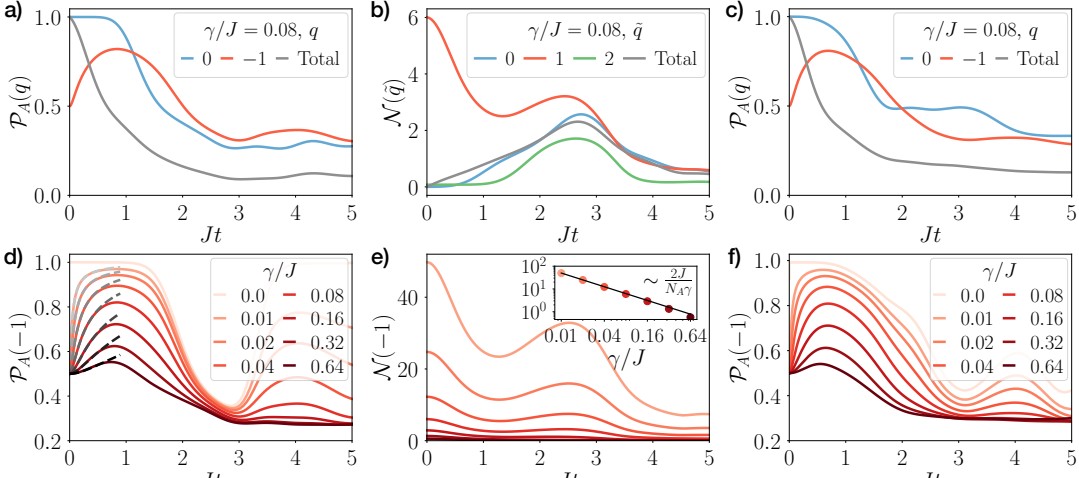

Figure 4: Dynamical purification and symmetry-resolved entanglement for one-dimensional XY spin models. We choose a system with $N = 8$, initialized in a Néel state $|\downarrow\uparrow\rangle^{\otimes N/2}$ and evolved with $H_{XY}$ subject to particle loss with rate $\gamma$ (see main text). We take $A = [1, 2, 3, 4]$ and $B = [5, 6, 7, 8]$. In panels (a,b,d,e), we consider short-range interactions $J_{ij} = J\delta_{i-1,j}$ and vanishing disorder $\delta/J = 0$. We present the symmetry-resolved purity $\mathcal{P}_A(q)$ [panels (a,d)] and the normalized symmetry-resolved negativity $\mathcal{N}(\tilde{q})$ [panels (b,e)] for various imbalance sectors $\tilde{q}$ (a,b) and decoherence rates $\gamma$ (d,e). The inset in (e) shows the early time value $\mathcal{N}(-1)|_{t=0^+}$ as function of $\gamma/J$. In panels (c,f), we present the symmetry-resolved purity for various imbalance sectors (c) and decoherence rates (f) in a system with long-range interactions $J_{ij} \sim J/|i - j|^{1.2}$ and a fixed disordered longitudinal field, sampled uniformly from $[-\delta, \delta]$ with $\delta/J = 0.86$. Gray lines in (d,e) are results from perturbation theory, Eqs. (46) and (32), respectively.

most strongly for weak decoherence [see Fig. 4d]. While the initial values $\mathcal{P}_A(1)|_{t=0^+} = 2/N_A$ is independent of $\gamma$, the peak of the purity is approaching the value of the purity for unitary dynamics. On the contrary, for $\gamma \gtrsim J$, the dynamics is dominated by decoherence, and purification is absent.

In Fig. 4 (b,e), we show the symmetry-resolved negativity $\mathcal{N}(\tilde{q})$. We observe that symmetry-resolved entanglement between $A$ and $B$ is dominated by the magnetization imbalance sector $\tilde{q} = -1$ sector. The magnitude of the negativity of sector $\tilde{q}$ is much larger than the total system negativity. In addition, as shown in the inset, the early time value at $Jt = 0^+$ is decreasing as $\sim 1/\gamma$ with increasing decoherence rate $\gamma$, as predicted by perturbation theory [Eq.(32)].

**Long-range, disordered spin chains. -** To illustrate the phenomenon of dynamical purification in more generic interacting spin chains, we display in panels (c,f) the symmetry-resolved purity in a system with long-range hopping with powerlaw coefficient $\alpha = 1.2$ and in the presence of a fixed disordered longitudinal field with $\delta = 0.86J$. Our findings are qualitatively very similar to the case of the (non-interacting) model with short-range interaction: dynamical purification is clearly observed in the symmetry sector $q = -1$ with increasing magnitude for weaker decoherence.

**Experimental setups. -** The dynamics discussed in this subsection is relevant for a variety of setups. In the next section, we will discuss and demonstrate implementation with trapped

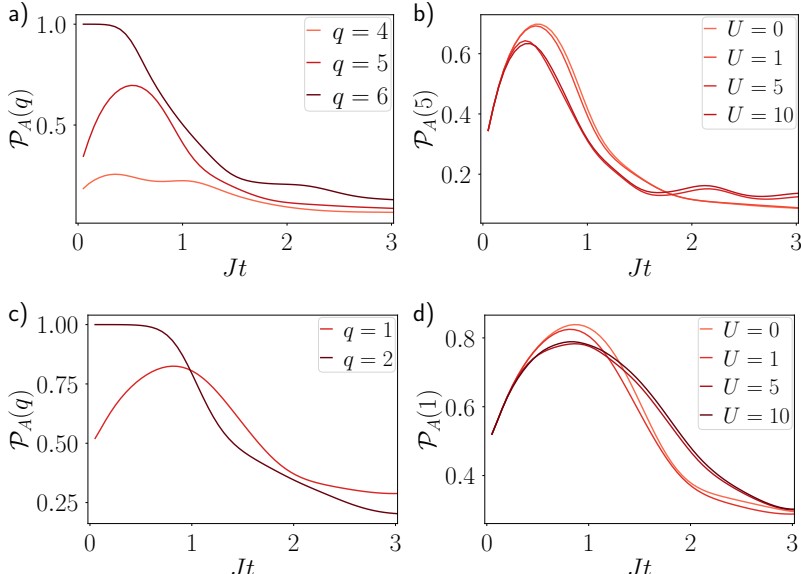

Figure 5: Symmetry-resolved purity for the Bose-Hubbard model in one dimension. We consider a system with $L = 8$ and we take $A = [1, 2, 3, 4]$ and $B = [5, 6, 7, 8]$. In panel a) and b) we start from the state $|\psi_0\rangle = |1, 2\rangle^{\otimes L/2}$ and consider: a) $U = 0.5J$, $\gamma = 0.1J$; b) and $q = 5$, $\gamma = 0.1J$. In panel c) and d) we start from the state $|\psi_0\rangle = |0, 1\rangle^{\otimes L/2}$ and consider: a) $U = 0.5J$, $\gamma = 0.1J$; b) and $q = 1$, $\gamma = 0.1J$.

ions in Paul traps. Another natural setting is Rydberg atoms in optical tweezers or optical lattices. Within those, the dipolar version of the XY Hamiltonian in Eq. (33) is naturally realized when considering direct dipole-dipole interactions within the Rydberg manifold (for a many-body demonstration, see Ref. [87]). Spin excitation losses occur naturally, and can be further enhanced via incoherently coupling the two Rydberg states. A very similar scenario (dipolar couplings) is also realized with superconducting qubits in 3D cavities, and with polar molecules or magnetic atoms in optical lattices.

## 4.2 Bose Hubbard model

In this section we discuss numerical simulations of the Bose-Hubbard model. This allows us to provide explicit evidence of dynamical purification in a full parameter regime connecting the strongly interacting case discussed above for the XY model, and Gaussian theories described later in the section.

The model Hamiltonian reads:

$$H_{BH} = J \sum_i \left( b_i^\dagger b_{i+1} + b_{i+1}^\dagger b_i \right) + U \sum_i n_i \left( n_i - 1 \right) . \tag{34}$$

Here $b, b^\dagger$ are bosonic operators, $n_i = b_i^\dagger b_i$ is the number operator on site $i$. For computational convenience, we truncate the number of bosons at a maximum of two per site (this also emulates well experiments in the presence of strong three-body losses [88,89]). The Hamiltonian preserves the total number of particles $N = \sum_i n_i$. The system dynamics is also subjected to particle loss modeled by $\gamma b_i$, $i = 1, \ldots, L$. The loss parameter $\gamma$ is fixed to $\gamma = 0.1J$ for all the simulations.

We consider a bipartite system of $L = 8$, where $A = [1, 2, 3, 4]$ and $B = [5, 6, 7, 8]$. According to the criteria discussed above, dynamical purification will take place for several choices of the initial state: we focus here on two cases: the state $|\psi_0\rangle = |1, 2\rangle^{\otimes L/2}$, where 2 means

that the site is doubly occupied while 1 means that there is a single boson; and the state $|\psi_0\rangle = |0,1\rangle^{\otimes L/2}$ where 0 denotes an empty site. We calculate, during the evolution of the system, the symmetry-resolved purity $\mathcal{P}_A(q)$, where $q$ denotes the number of particles in subsystem $A$.

In Fig. 5 we plot the symmetry-resolved purity as a function of time, starting from the state $(1,2,1,2..)$: in panel a) we consider sectors $q = 4,5,6$ with $U = 0.5J$, in panel b) we fix $q = 5$ and take into account different values of $U$. In agreement with our theory, we observe no purification in the sector $q = 6$ in panel a) since it is the only one occupied in the initial state. Sectors $q = 4,5$ instead purify at intermediate times with a more pronounced purification visible in the nearest sector $q = 5$. Afterwards the curves approach the same value of purity as information equipartition shall occur at long times. In panel b) we observe the same phenomenology for several values of the interaction strength. Here the sector $q = 5$ experience purification but the value and the position of the peak changes as function of $U$.

In panels c) and d) of Fig. 5, we investigate the same scenario but starting from the state $(0,1,0,1..)$, and consider in panel c) the symmetry-resolved purity for sectors $q = 1,2$ with $U = 0.5J$, and in panel d) the same quantity for $q = 1$ and several values of $U$. We observe the same phenomenology of panel c) and d). Dynamical purification is, in fact, present for any initial state which is a product state. Here we observe that the sector $q = 1$ purifies and the effect is present also when the strength of the interaction increases, as witnessed in panel d).

The results of the Bose-Hubbard and XY models indicate, as predicted by our theory, that dynamical purification occurs over the entire interaction regime - from weak to infinite coupling. It is worth noticing that the maximum purity is weakly affected, while the time to reach the maximum itself is sensitive to both initial filling fraction, and interactions. The first effect can be traced back to bosonic enhancement. The second is instead likely due to the effect of a more constrained dynamics for strong interactions (many states becoming non-resonant), that is likely affecting terms beyond second order in perturbation theory.

**Experimental setups. -** Bose-Hubbard models with single particle losses describe well the dynamics of cold atoms in optical lattices, where some of the probing techniques introduced here can be implemented [31, 33].

### 4.3 U(1) lattice gauge theory

An even stronger form of interacting system in 1D is provided by gauge theories with U(1) center. For those models, Coulomb interactions follow a genuine linear increase as a function of distance due to confinement. However, since charge creation is still a local process, one expects dynamical purification to still occur, albeit at a possibly slower rate when compare to models with local interactions. In order to illustrate this, we have investigated the short time dynamics of the lattice Schwinger model, a U(1) lattice gauge theory describing the coupling between fermions and U(1) gauge fields. The model Hamiltonian reads:

$$H = w \sum_i (\psi_i^\dagger U_{i,i+1} \psi_{i+1} + \text{h.c.}) + J \sum_i E_i^2 + m \sum_i (-1)^i \psi_i^\dagger \psi_i , \qquad (35)$$

where $\psi$ are fermionic annihilation operators, $U$ are U(1) parallel transporters, and $E$ is corresponding electric field terms. The first term describes minimal coupling, the second the field interaction strength, and the third represents a mass term, that features a staggering typical of Kogut-Sussking (also known as staggered) fermions. For our simulations, we find it convenient to recast the Schwinger model as a spin Hamiltonian with long range interaction in

the following way [90]:

$$H_S = H_\pm + H_Z + H_E\,,$$
$$H_\pm = w \sum_i \left(\sigma_i^+ \sigma_{i+1}^- + \sigma_i^- \sigma_{i+1}^+\right),$$
$$H_Z = \frac{m}{2}\sum_i (-1)^i \sigma_i^z - \frac{J}{2}\sum_{n=1}^{N-1}(n\,\mathrm{mod}\,2)\sum_{l=1}^{N}\sigma_l^z\,,$$
$$H_E = J\sum_{n=1}^{N-1} E_n^2\,.$$

(36)

Here $\sigma_j^\pm = (\sigma_j^x \pm i\sigma_j^y)/2$. It can be shown that

$$H_E = J\sum_{n=1}^{N-1}\left[\epsilon_0 + \frac{1}{2}\sum_{l=1}^{n}\left(\sigma_l^z + (-1)^l\right)\right]^2.$$

(37)

It gives rise to a long range spin-spin interaction and local energy offsets, that represent Coulomb law between staggered fermions.

In Fig. 6 we plot the symmetry-resolved purity during the dynamics of spin system evolving under the Hamiltonian in Eq. 36, starting from the state $|\Psi_0\rangle = |\downarrow\uparrow\rangle^{\otimes N/2}$ with total magnetization $S_z = \sum_{i=1}^{N}\sigma_i^z = 0$. We observe that the coherent dynamics preserves the magnetization while the dissipation, modeled by $\sqrt{\gamma}\sigma_i^-$ ($\forall i = 1,\dots,L$), does not.

In panel a) the symmetry resolved purity is plotted for $w = 1, \epsilon_0 = 0$, $m = 0$, $J = 0.1$, $\gamma = 0.05$. In panel b) we fix $w = 1$, $m = 0$, $\gamma = 0.05$ and let $J$ and $\epsilon_0$ to vary. We see that even a long-range, strongly interacting system experience dynamical purification in both sectors $q = -2, -1$. Changing the values of the long range coupling $J$ and of the background field $\epsilon_0$ does not change qualitatively the picture. A richer structure seems to emerge in the $q = -2$ sector at long times, suggesting that, while dynamical purification occurs smoothly, entanglement equipartition does not.

**Experimental setups. -** The dynamics of the Schwinger model in the Wilson formulation has been realized in 4-site trapped ion experiments [91]. However, one would need larger system sizes to observe dynamical purification. Several proposals, based on a variety of platform, exist [92], either in the formulation including gauge fields, or on the integrated theory. Specific dissipation sources have not been discussed in detail so far: however, in most platforms, they are likely to be similar to the Bose-Hubbard case discussed above [92, 93].

### 4.4 Fermionic systems in 1D and 2D

While all models discussed so far are intrinsically interacting, we now provide numerical evidences of the physics described in the previous sections in free fermionic systems [94, 95]. The latter allow us to consider larger system sizes and two-dimensional geometries. Most importantly, it allows us to check systematically specific features of our predictions, such as the dependence on the partition size, dimensionality, and topology of the partition (e.g.: in 1D, we will consider explicitly disconnected partitions).

We start from a charge-density-wave (half-filling), and let it evolve according to a GKSL master equation master with jump operator $l_j = \gamma c_j$ and Hamiltonian:

$$H = -J\sum_{\langle i,j\rangle} c_i^\dagger c_j - 2\mu\sum_j\left(c_j^\dagger c_j - \frac{1}{2}\right).$$

(38)

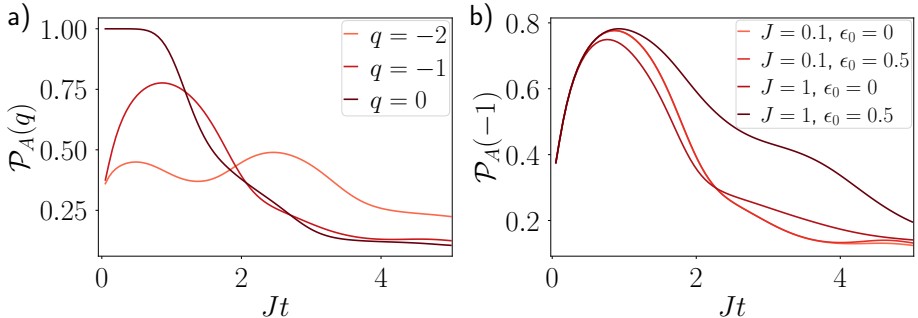

Figure 6: Symmetry-resolved purity for the Schwinger model in one dimension. We consider a system with $L = 12$ and we take $A = [1, 2, 3, 4, 5, 6]$ and $B = [7, 8, 9, 10, 11, 12]$. Panel a): $w = 1$, $\epsilon_0 = 0$, $m = 0$, $J = 0.1$, $\gamma = 0.05$. Panel b) $w = 1$, $m = 0$, $\gamma = 0.05$. The dynamics starts from state $|\downarrow\uparrow\rangle^{\otimes N/2}$.

The first sum runs over nearest neighbours, $c_i^\dagger, c_i$ denote fermionic creation/annihilation operators, $J$ is the hopping constant (that we set to unity below, $J = 1$) and $\mu$ is the chemical potential ($\mu = 0$ unless stated otherwise). In free fermionic theories, at each time $t$ one can compute the charged-moments $Z_n(\alpha)$ (Eq. (7)) via the two-point fermionic correlation matrix $C_{ij} = \langle c_i^\dagger c_j \rangle$ and its evolution according to Ref. [96]. We consider both 1$D$ chains and 2$D$ square lattices and check numerically the analytical predictions in the previous sections. In 1D, the tight-binding model is mapped to the XY Hamiltonian (33) by a Jordan-Wigner transformation (but the jump operators are different): the GKSL master equation we will consider are similar to one of the examples discussed in Ref. [97].

In Fig. 7, we show some representative numerical results. In panels a)-b) we consider $\mathcal{P}_A(q)$, cf. Eq. (5), in 1D. The system is divided into three parts as $\mathcal{S} = A \cup B \cup A$ with $|A| = \ell/2$ and $\ell = L/2$, a representation of the system is in Fig 7a). The choice of the topology of the partition allows us to illustrate the generality of dynamical purification, that is indeed topology independent as long as $\ell \gg tJ$. In panel c) we compute the same quantity for a two-dimensional square lattice to highlight that the features of the dynamics are not dependent on the dimensionality or connectivity of the partition. Here we consider $\mathcal{S} = A \cup B$ where $A$ is a square of linear dimension $\ell = L/4$ at the center of the system. In panels d)-e)-f) we focus on the behavior of the symmetry-resolved purity in the absence of dissipation, to emphasize that the bath plays a decisive role in the dynamical purification, and on quenches starting from different states, since we expect our results to hold when the initial state is separable (cf. 3.1). The initial state being at half-filling, $q = \ell/2$ is the only populated sector at $t = 0$. We will consider the quantity $\mathcal{P}_A(q)$ where, for instance, $q = 1$ refers to the sector $\ell/2 + 1$ (one particle more than half-filling). We omit $\ell/2$ to be concise. Let us now discuss the plots in details. In all the following simulations we always consider open boundary conditions (OBC).

In Fig. 7a), we show $\mathcal{P}_A(q)$ for $q = 0, 1, 2, 3$, $L = 128$. The sector $q = 0$ is the only one occupied at $t = 0$. It is pure at the start of the evolution and does not experience any purification. Oppositely, as soon as dynamics kicks in, the sector $q = 1$ becomes mixed. Its purity increases at intermediate times (*dynamical purification*) and approaches *equipartition* for longer times. This is highlighted in the inset showing the behavior of $\mathcal{P}_A(q)$ for $Jt \in [1, 5]$ in logarithmic scale. The purification for the other sectors is present, but less evident as it is connected to higher-order perturbative processes.

In Fig. 7b) we fix $q = 1$ and consider $\mathcal{P}_A(q)$ for different values of $\ell$ with $L = 2\ell$. In agreement with theory, the peak of the curves decreases, approaching zero. The point at $t = 0^+$ should approach zero as $\sim 1/\ell$, as well, like it has been anticipated in the previous sections. The inset shows a fit of $\mathcal{P}_A(q = 1, t = 0^+)$ as a function of $\ell$, which demonstrates the

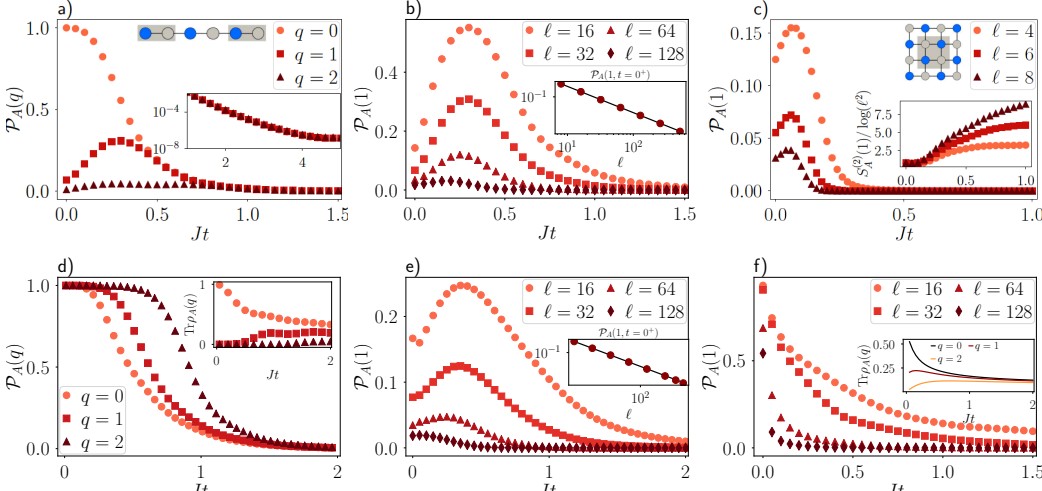

Figure 7: Results of the simulation of $\mathcal{P}_A(\ell/2 + q)$ for a quadratic open fermionic system. We omit $\ell/2$ and use $q = q - \ell/2$ to label the symmetry sectors. Parameters: $J = 1$, $\mu = 0$. First line: symmetry-resolved Rényi entropy for a) 1D system with $L = 64$, $l = 32$, $\gamma = 0.05$; b) 1D system for $L = 2\ell$, $q = 1$, $\gamma = 0.05$; c) 2D system with $L = 4\ell$, $N = L^2$, $q = 1$, $\gamma = 0.2$. Second line: symmetry-resolved Rényi entropy for d) 1D system with $L = 64$, $l = 16$, $\gamma = 0$, purely coherent dynamics; e) 1D system for $L = 2\ell$, $q = 1$, $\gamma = 0.05$, starting from the Majumdar-Ghosh dimer state; f)1D system for $L = 2\ell$, $q = 1$, $\gamma = 0.05$, starting from the ground state of a nearest-neighbours tight binding model with $J = 1$.

log-volume regime already discussed.

The behavior of the symmetry-resolved purity for a two-dimensional systems is analogous. In Fig. 7c) we plot the purity, at fixed $q = 1$, for different values of $L$. The total number of sites of the lattice is $N = L^2$ and the subsystem $A$ consists of $l^2 = N/16$ sites picked at the center of the square. Studying the position of the point at $t = 0^+$ one observes that it scales as $\sim 1/\ell^2$ as calculated in Eq. (22) and shown in Fig. 2a): this confirms a 2D log-volume scaling at short times, with the corresponding symmetry-resolved Rényi entropy displayed in the inset for the sake of completeness.

In Fig. 7d), we take into account the symmetry-resolved purity in the case of a purely coherent dynamics. This show remarkably how the purification process is strictly related to the presence of a bath for this class of models. We consider $L = 64$ and $\ell = 32$. While $q \neq 0$ sectors are mixed at time $t = 0^+$ in presence of bath, this is not the case for $\gamma = 0$. In the inset one can see the population of each given sector as a function of time. As the coherent dynamics starts playing its role, the population increases and the purity decreases correspondingly. The $q \neq 0$ sectors are involved in the evolution but they do not experience any purification, instead their purity decreases monotonously to a unique value independent of $q$, witnessing information equipartition. All these results are compatible with the exact ones reported in Ref. [12].

Finally, in Fig. 7e-f), we depict the symmetry-resolved purity in the sector $q = 1$, in the case of a global quench starting from two different states. Firstly, in e), we consider a global quench from the Majumdar-Ghosh dimer product state and an evolution under the Hamiltonian in Eq. (38); secondly, in f), we take as starting point the ground state of Eq. (38) and evolve the

system according to a long range hopping Hamiltonian in the form:

$$H = -\sum_{ij} \frac{J}{|i-j|^\alpha} c_i^\dagger c_j, \tag{39}$$

where $\alpha = 2$.

The purpose of panels e) and f) is to show that the dynamical purification is present only in the case the initial state is separable, as it happens for Fig. 7b)-e). In the inset of Fig. 7e) we show a fit of $\mathcal{P}_A(q=1, t=0^+)$ as a function of $\ell$, which exhibits a $1/\ell$ behavior, as predicted by perturbation theory. Oppositely, if the initial state is entangled, one cannot see any emergent purification during the dynamics (Fig. 7f)). This is due to the fact that the symmetry-resolved reduced density matrix is already mixed in all $E_k$ sectors, and thus, local coherent dynamics is insufficient to purify the state, as the number of non-zero eigenvalues in each sector is exponentially large in the partition size. In the inset of the figure the populations of sectors $q = 0, 1, 2$ for $L = 256$ are shown. Evidently, all the sectors are occupied already at $t = 0$.

**Experimental setups.** - The $U(1)$ dynamics discussed in this section is of direct relevance for various experimental settings. The first ones are fermionic or (hard-core) bosonic atoms trapped into optical lattices. There, one of the main sources of dissipation (in addition to spontaneous emission, that can be made small with the use of blue detuned lattices) is single particle loss. While in principle loss rates due to inelastic background scattering are small when compared to the typical lattice dynamics, localized losses can be engineered in a variety of ways, including weak-laser coupling to untrapped levels or via electron beams.

The second setting that is relevant to this subsection are arrays of superconducting qubits. In the strong coupling limit, the dynamics of such systems can be well approximated by an XY model. Qubit relaxation will then play the same role as single particle loss.

# 5  Experimental protocol for measuring symmetry-resolved purities

Our protocol to extract symmetry-resolved purities is based on randomized measurements. These methods have been introduced and experimentally demonstrated to measure entanglement entropies [29, 32, 33, 35], and other nonlinear functions of the density matrix, such as state fidelities [98], out-of-time ordered correlators [99, 100], topological invariants [101, 102], and entanglement negativity [37, 103]. In the quantum information context, the moments of statistical correlations between randomized measurements can also be used to define powerful entanglement witnesses without reference frames [37, 104–108].

While standard projective measurements performed in a fixed basis can only give access to expectation values of a particular observable, randomized measurements consist instead in measuring our quantum state in different random bases, giving access to complicated nonlinear functionals of the density matrix, here symmetry-resolved purities.

As in Refs. [35, 37], our approach is based on the idea of combining two results: randomized measurement tomography [31, 109], and 'shadow' tomography [35, 110]. Let us consider here a spin system and show how to measure the symmetry-resolved purity of a reduced state $\rho_A$ made of $N_A$ spins.

Randomized measurements are realized by applying random local unitaries $\rho_A \to u \rho_A u^\dagger$, $u = u_1 \otimes \cdots \otimes u_{N_A}$, where each $u_i$ is a spin rotation that is taken, independently, from a unitary 3-design [111, 112]. After the application of random unitary, a projective measurement is realized in a fixed basis. This procedure is repeated with $M$ different random unitaries, in order to obtain a list of $M$ measured bitstrings $\mathbf{k}^{(r)}$, $r = 1, \ldots, M$.

Randomized measurementsare tomographically complete in expectation and can be used to provide an estimator of the density matrix [31, 35, 36, 109, 113], a classical shadow [35],

$$\hat{\rho}_A^{(r)} = \bigotimes_{i \in A} \left[ 3(u_i^{(r)})^\dagger \left| k_i^{(r)} \right\rangle \left\langle k_i^{(r)} \right| u_i^{(r)} - \mathbb{I}_2 \right], \tag{40}$$

with the expectation value over randomized measurements $\mathbb{E}[\hat{\rho}_A^{(r)}] = \rho_A$. It is not our aim to reconstruct the density matrix based on Eq. (40) i.e., to perform tomography, as it will be too costly in terms of measurements (and classical post-processing). However, we can make use of this expression Eq. (40), in order to relate directly *polynomial functionals* of $\rho$ to the measured data $\mathbf{k}^{(r)}$ [35]. For the symmetry-resolved purity, simply consider two independent randomized measurements $r \neq r'$, and define the symmetrized estimator

$$\mathcal{P}_A(q)^{(r,r')} = \tfrac{1}{2}\text{tr}[(\hat{\rho}_A^{(r)}\Pi_q)(\hat{\rho}_A^{(r')}\Pi_q)] + \tfrac{1}{2}\text{tr}[(\hat{\rho}_A^{(r')}\Pi_q)(\hat{\rho}_A^{(r)}\Pi_q)]. \tag{41}$$

Using Eq. (40), this can be seen as a simple bi-linear function of the measurement data. Averaging over many pairs $(r, r')$, boosts convergence to the estimator's expectation value

$$\mathbb{E}[\mathcal{P}_A(q)^{(r,r')}] = \tfrac{1}{2}\text{tr}[(\mathbb{E}[\hat{\rho}_A^{(r)}]\Pi_q)(\mathbb{E}[\hat{\rho}_A^{(r')}]\Pi_q)] + \tfrac{1}{2}\text{tr}[(\mathbb{E}[\hat{\rho}_A^{(r')}]\Pi_q)(\mathbb{E}[\hat{\rho}_A^{(r)}]\Pi_q)] = \mathcal{P}_A(q).$$

Here, we have used that $\hat{\rho}_A^{(r)}$ and $\hat{\rho}_A^{(r')}$ are independent realizations of Eq. (40). This means that $\mathcal{P}_A(q)^{(r,r')}$ is an unbiased estimator of the symmetry-resolved purity. This procedure can be straightforwardly extended to higher moments with triplets of randomized measurements $r \neq r' \neq r''$, etc. Appropriate implementation of partial transposes moreover allows for extracting symmetry-resolved Rényi entropies (6). This is the content of Ref. [30], where we also provide a thorough statistical analysis for estimating symmetry-resolved quantities based on randomized measurements. The upshot is that estimator (41) can be equipped with rigorous confidence bounds. Already $2^{N_A}\mathcal{P}_A(q)/\epsilon^2$ measurement repetitions suffice to estimate a given symmetry-resolved purity $\mathcal{P}_A(q)$ up to accuracy $\epsilon$. This favorable scaling is a key advantage over full quantum state tomography In particular, the scaling depends only on the symmetry resolved purity $\mathcal{P}_A(q)$ and not on (the inverse) of the population $\text{tr}\{\rho_A\Pi_q\}$. This can make a large difference, especially when the population is tiny. Our analysis of experimental data, c.f. next section, support this favorable picture.

Therefore, we believe that symmetry-resolved purities can be measured in various NISQ platforms up to moderate partition sizes $N_A = 10 \sim 20$, which are sufficient large to observe many-body effects, such as dynamical purification. The second advantage of randomized measurements with respect to tomographic-type estimations is the post-processing step. Here, the estimation of $\mathcal{P}_A(q)^{(r,r')}$ from the data simply consists in multiplying estimators $\hat{\rho}_A^{(r)}$ with an efficient tensor-product representation (Eq. (40)) with a projection operator with sparse-matrix structure (which can be for instance efficiently written as a Matrix-Product-Operator [114]).

Note finally that here randomized unitaries do not have a symmetric structure, and therefore each estimation of the density matrix does not have a block-diagonal structure. Alternatively, one can envision to perform symmetry-resolved random unitaries incorporating symmetries [31, 33, 34]. While these random unitaries appear as more challenging to realize experimentally compared to local spin rotations, one should expect a reduction of statistical errors in this situation [109].

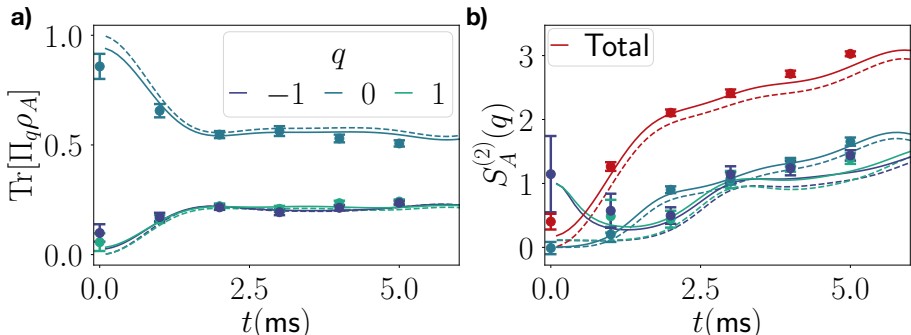

Figure 8: Experimental demonstration of symmetry-resolved purification in a trapped ion quantum simulator, using data obtained in the context of Ref. [29]. We consider a system of $N = 10$ spins, with subsystems $A = [4, 5, 6, 7]$ and $B = [1, 2, 3, 8, 9, 10]$. In panels a) and b), the symmetry-resolved populations and Rényi entropies of various magnetization sectors $q = 0, \pm1$ of the reduced density matrix $\rho_A$ are shown as function of time (see Fig. 2 for symmetry-resolved purities). Error bars have been calculated with Jackknife resampling. In panel b), data for the magnetization sector $q = 1$ at $Jt = 0$ has been omitted due to large errorbars, resulting from small populations. Lines are numerical simulations of unitary dynamics (dashed) and including decoherence (solid) as decribed in the text.

# 6 Experimental observation of dynamical purification in trapped ion chains

In this section, we demonstrate symmetry-resolved purification experimentally in a trapped ion quantum simulator, using data taken in the context of Ref. [29]. Here, quench dynamics with a long-range XY-model introduced in Eq. (33), with $\delta_i = B$, $J_{ij} \approx J/|i - j|^\alpha$ the coupling matrix with an approximate power-law decay $\alpha \approx 1.24$, and $J = 420s^{-1}$. The effective magnetic field is taken to be large $B \approx 2\pi \cdot 1.5\text{kHz} \approx 22J$: this way, that the unitary dynamics conserve the total magnetization $S_z = \sum_i \sigma_i^z$, since terms that would break it (such as $\sigma_i^+ \sigma_j^+ + \text{h.c.}$) are energetically suppressed [29].

In addition, decoherence is present in the experiment, during initial state preparation, time evolution and the randomized measurement. As detailed in Ref. [29], we can model these decoherence effects as follows.

The time evolution is subject to local spin-flips, and spin excitation loss (spontaneous decay). We describe the corresponding dynamics with a master equation with jump operators $C_i = \sqrt{\gamma_F}\sigma_i^x$ for $i = 1, \dots, N$ and $C_{i+N} = \sqrt{\gamma_D}\sigma_i^-$ for $i = 1, \dots, N$, capturing the spin flip and excitation loss, respectively. Here, the rates are $\gamma_F \approx \gamma_D \approx 0.7/s$.

In the experiments, the initial state is not pure, but rather it is a mixed product state $\rho_0 = \bigotimes_i (p_i |\uparrow\rangle \langle\uparrow| + (1 - p_i)|\downarrow\rangle \langle\downarrow|)$ with $p_i \approx 0.004$ for $i$ even and $p_i \approx 0.995$ for $i$ odd. Finally, during the application of the local random unitary, local depolarizing noise is acting which is modeled as

$$\rho(t_{\text{final}}) \rightarrow (1 - p_{DP}N)\rho(t_{\text{final}}) + p_{DP}\sum_i \text{Tr}_i[\rho(t_{\text{final}})] \otimes \frac{\mathbb{1}_i}{2}, \tag{42}$$

with $p_{DP} \approx 0.02$.

In Fig. 8, we present experimental results, obtained with the estimators defined in Eq. (41), and numerical simulations, for unitary dynamics and including the decoherence model described above. In panel a), the populations $\text{tr}\{\Pi_q\rho_A\}$ of the magnetization sectors $q$ of the

reduced density matrix $\rho_A$ are shown, with $A$ consisting of spins $A = [4, 5, 6, 7]$, and $\mathcal{V}_A = 4$. Initially, the $(q = 0)$-sector is predominantly populated, with small fractions in other sectors, due to the finite initial state preparation fidelity. With time, the population in other sectors, in particular $q = \pm 1$, increases.

The symmetry-resolved second Rényi entropy $S_A^{(2)}(q)$ is shown in panel b) for various magnetization sectors (see Fig. 2 for the corresponding symmetry-resolved purity). The experimental data clearly shows dynamical purification (decrease of the Rényi entropy) in the $q = -1$ sector. Data in the $q = +1$ sector are also suggestive of dynamical purification, even if a strong statement cannot me made here do to comparatively larger error bars. In particular, this demonstrates that dynamical purification can be observed in one-dimensional systems with algebraically decaying long-range interactions (see also Sec. 3.1). At long times, the symmetry-resolved Rényi entropies approach similar values for all displayed sectors, consistent with expected equipartition of the symmetry sectors [10]. Finally, we note that, in the experiment, there is clear separation of scales $t_J \ll t_2$, while $t_1$ and $t_J$ are not separated. As discussed in the theory section, this shows compellingly how the second condition is not required to observe dynamical purification, since at short times, decoherence dominates regardless of the volume of the partition considered.

## 7 Conclusions

Symmetry is an ubiquitous element characterising synthetic quantum matter - from quantum simulators, to noisy-intermediate scale quantum devices. In this work, we have developed a theoretical framework for the description of symmetry-resolved information spreading in such open quantum systems, focusing on the epitome case of $U(1)$ symmetries common to several experimental platforms - from cold gases in optical lattices, to trapped ions and superconducting circuits. We have shown how, for various settings encompassing a wide spectrum of interacting and non-interacting theories, specific quantum number sectors undergo dynamical purification under ubiquitous conditions of weak noise and separable initial states, without experiencing quantum information loss. Such phenomenology is general, occurs in any dimension, is not sensitive to the partition topology, and features specific scaling scenarios for the entropy as a function of partition size. Most importantly, the dynamical purification considered here occurs in symmetric systems and stems from the competition between coherent and incoherent dynamics that is a leitmotif of current NISQ devices.

We have introduced and experimentally demonstrated a protocol to measure symmetry-resolved quantum information quantities based on a combination of randomized measurement probing and shadow tomography. Our approach is scalable to partition sizes that are well beyond what is accessible to full state tomography, and is applicable to a broad spectrum of experimental settings with single site control and high repetition rate. Both scalability and applicability are of key importance in order to probe genuinely many-body features of entanglement dynamics in state-of-the-art experiments. Two key features of our protocol are the fact that symmetry can be enforced *a posteriori* on a given data set, without necessarily relying on the implementation of symmetry-preserving random unitaries, and that errors are provably under control even in cases where populations in given subsectors are small (that is a challenge specific to symmetry-resolved density matrices). Based on our protocol, we have shown how the experiments performed in Ref. [29] have already realized dynamical purification in a trapped ion chain described by a long-range XY model. This observation, in full agreement with our theory predictions, testifies for the generality of symmetry-resolved dynamical purification under experimentally realistic conditions. While our protocol is generically applicable to lattice models, it would be interesting to extend it to continuous systems, where the role

of symmetry-resolved information is relatively unexplored outside of conformal field theories [83, 115].

The capability of addressing the combined role of symmetry and quantum correlations in NISQ devices opens a novel interface between theory and experiments, where many-body effects intertwine with information theoretic applications. The first instance of that is what role symmetry plays in quantum information protocols, in particular, error correction. Our tools may be of particular importance here, as several error correcting codes can be cast as gauge theories, one example being the toric code [116]. In this context, the role of specific symmetry sectors is associated to the presence of excitations. It may thus be useful to employ the experimental tools we have used here to access how specific perturbations compromise the reliability of a quantum memory. Going beyond that, understanding whether dynamical purification occurs in the presence of local symmetries is an open question, that could be in principle addressed within the same methods presented here.

The second possible applications of our methods concerns the capability of utilizing dynamical purification as a proxy of the system dynamics, in particular, to determine its dissipative dynamics. One first element is that dynamical purification is expected for a quantum noise, that is local: it is thus informative about the nature of the dissipation. The fact that the dissipation rates intertwines with the partition size could also help to quantify the relative strength of incoherent versus coherent processes, at least in cases where specific initial states could be realized with high fidelity. Remarkably, despite being a short-to-intermediate time phenomenon, thanks to the area-to-volume ratio being tunable, dynamical purification is also informative about very weak dissipation: this is particularly important for diagnostics, as one would expect that the latter requires long-time evolution to be characterized.

On more general grounds, symmetry-resolved dynamical purification reveals how certain many-body phenomena can only be properly characterized utilizing symmetry to emphasize or even magnify relevant information. In particular, symmetry-resolution allows to properly diagnose physical phenomena that would not be accessible otherwise, by amplifying the role of sectors in the reduced density matrix whose information content could be otherwise overwhelmed by other less informative - but highly-weighted - sectors. In this context, the many-body theory we develop seems to suggest that symmetry can be used to develop improved entanglement detection that could outperform their respective 'symmetry-blind' counterparts [30].

# Acknowledgements

We acknowledge useful discussions with L. Capizzi, R. Fazio, and S. Murciano. We thank T. Brydges, P. Jurcevic, C. Maier, B. Lanyon, R. Blatt, and C. Roos for generously sharing the experimental data of Ref. [29].

**Funding information**    This work is partly supported by the ERC under grant number 758329 (AGEnTh) and 771536 (NEMO), and by the MIUR Programme FARE (MEPH). MD, AE, VV and PZ acknowledge support from the European Union's Horizon 2020 research and innovation programme under grant agreement No 817482 (Pasquans). AE and PZ acknowledge funding by the European Union program Horizon 2020 under grant agreement No. 731473 (QuantERA via QTFLAG), the US Air Force Office of Scientific Research (AFOSR) via IOE Grant No. FA9550-19-1-7044 LASCEM, by the Simons Collaboration on UltraQuantum Matter, which is a grant from the Simons Foundation (651440, PZ), and by the Institut für Quanteninformation. Work in Trieste has been carried out within the activities of TQT. BV acknowledges financial support from the Austrian Science Fundation (FWF, P. 32597N), and the French National

Research Agency (ANR, JCJC project QRand). J. C., A. N. and B. K. acknowledge financial support from the Austrian Science Fund (FWF) stand alone project: P32273-N27 and the SFB BeyondC: F 7107-N38.

# A  Effective Markovian dynamics for the symmetry-resolved reduced density matrix

We now provide a simple interpretation of the effective description derived in Sec 3. Our main interest here is to determine whether dynamical purification is an effect that relies on a specific correlation present in an effective bath (derived by applying the symmetry-resolved projectors to the density matrix), or whether it is unrelated to that, and thus captured entirely by an emergent Markovian dynamics describing $\rho_{A,q}$.

Indeed, even though the evolution of the global density matrix $\rho$ is governed by the Markovian master equation of Eq. (18), the symmetry-resolved reduced density matrix $\rho_{A,q}$ could have a non-Markovian time evolution. The dissipation rates derived in Eq. (19) can be interpreted as an effective master equation acting directly on the symmetry-resolved reduced density matrix, with time dependent rates. We consider two arbitrary density-matrices product states, whose symmetry-resolved reduced density matrix can be written in diagonal form $\rho_I, \rho_{II}$, with matrix elements $a_{jj;I}$ and $a_{jj;II}$, respectively. What we are interested in is whether the two states can become dynamically more distinguishable as a function of time: if this is possible even for a finite time window, the time evolution is non-Markovian [117]. In order to address this point, we define the distance between these states as:

$$D_{I,II} = \text{Tr} \sqrt{\rho_I \rho_{II}}. \tag{43}$$

After a few lines of algebra, and defining as $N, M$ the total rank of the density matrix and the number of the states belonging to $E_0$, respectively, one obtains:

$$
\begin{aligned}
\frac{\partial D}{\partial t} = & -\frac{1}{2(1+N\gamma t+MJ^2 t^2)^2} \\
& \times \left[ \sum_{j\in E_0} \frac{\gamma(a_{jj;I}+a_{jj;II}+2\gamma t)}{\sqrt{(a_{jj;I}+\gamma t)(a_{jj;II}+\gamma t)}} + \sum_{j\in E_1} \frac{(\gamma+2J^2 t)(a_{jj;I}+a_{jj;II}+2\gamma t+2J^2 t^2)}{\sqrt{(a_{jj;I}+\gamma t+J^2 t^2)(a_{jj;II}+\gamma t+J^2 t^2)}} \right] \\
& -\frac{2(N\gamma+2MJ^2 t)}{(1+N\gamma t+MJ^2 t^2)^3} \\
& \times \left[ \sum_{j\in E_0} \sqrt{(a_{jj;I}+\gamma t)(a_{jj;II}+\gamma t)} + \sum_{j\in E_1} \sqrt{(a_{jj;I}+\gamma t+J^2 t^2)(a_{jj;II}+\gamma t+J^2 t^2)} \right],
\end{aligned}
$$

so that, under the condition above, one has $\partial D/\partial t < 0$, as this is just the sum of two negative terms. This implies that arbitrary states of the type discussed above become less distinguishable as a function of time, a signature of effective Markovian dynamics.

The very same conclusion can be obtained on more general grounds by noticing that the rate equations above all have positive rates, thus satisfying P-divisibility criteria. At the physical level, this is a consequence of the fact that the relaxation time of the environment (in this case, the part of the system we are tracing upon in a symmetry-resolved fashion [8]) is much longer than the timescales we are interested in. For longer times (not accessible to the

---

[8]We emphasize that we are dealing with a specific symmetry-resolved sector, as other sectors may feature information backflow - a proxy of non-Markovianity - at earlier times.

regime we can tackle with our theory, but definitely numerically accessible), we expect that such effective Markovian description would ultimately break down due to the bath dynamics timescales being comparable to the one characterizing the partition.

## B Symmetry-resolved purification in one-dimensional systems

The system we consider here is a one-dimensional version of the system considered in Sec. 3.1, that we divide into two connected partitions $A \cup B$ with $N_A$ and $N_B$ sites, respectively. We assume that the system is initialized in a charge-density wave $|\psi_0\rangle = |\downarrow, \uparrow, \ldots, \uparrow\rangle$ (a Néel state), and for simplicity, take $N_A$ and $N_B$ to be even. We consider dynamics governed by Eq. (24) and focus on timescales accessible within perturbation theory, that is, $J^2 t^2, t\gamma \ll 1$. We are interested in the sector $q = -1$. Adapting the 2D calculations presented in the main text, we find that $\rho_A(q = -1)$ is divided into two blocks (in perturbation theory):

1. $E_0(-1)$: the state that is connected to the CDW by a single hopping process, or a loss event, at the boundary with rate $\lambda_0^{E_0}$

2. $E_1(-1)$: the $(N_A/2 - 1)$ states that are connected to the CDW by a single loss in the rest of the system with eigenvalue $\lambda_k^{E_1}$;

At the lowest order in perturbation theory, one has the following scaling of the eigenvalues of $\rho_A(-1)$:

$$\lambda_0^{E_0} = (J^2 t^2 + \gamma t)/A(t), \; \lambda_k^{E_1} = \gamma t/A(t), \tag{44}$$

with normalization

$$A(t) = \gamma t (N_A/2) + J^2 t^2. \tag{45}$$

This gives

$$\mathcal{P}_A(-1) = \frac{(N_A/2 - 1)\gamma^2 t^2 + (J^2 t^2 + \gamma t)^2}{[(N_A/2)\gamma t + J^2 t^2]^2}. \tag{46}$$

## C Sectors populations

In the quench dynamics we investigate, the population in each different subsector plays an important, practical role: it quantifies how relevant is the sector where dynamical purification occurs. Within the Bose-Hubbard model example, considering the approximation that the $(-1)$ and $(0)$ sectors are the only ones populated at short times, one expects that the population of the latter increases as a function of time in a manner that is linear at very short times, and quadratic in the regime of purification (a scaling similar to the prefactor $A(t)$).

In Fig. 9, 10, 11, we show some sample results that illustrate this fact (that is also found in the experimental data). It is interesting to note that, for most cases, the population in the $(-1)$ sectors is comparable to the one in the $(0)$ around the purification timescale, irrespectively of system size and interaction regime. This feature signals that observing such effect beyond the spin models discussed in the text should involve only very modest post-selection.

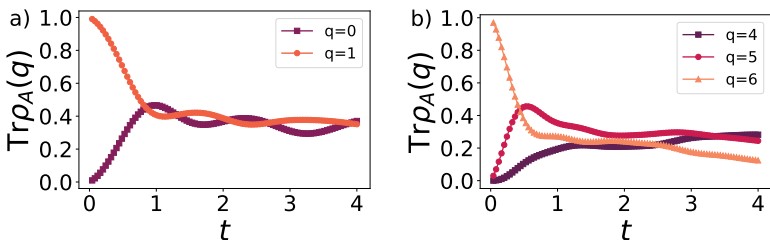

Figure 9: Population of the symmetry sectors during the evolution of the systems described in Fig.s 5. We consider a system with $L = 8$ and we take $A = [1, 2, 3, 4]$ and $B = [5, 6, 7, 8]$ Here we fix $U = 0.5J$, $\gamma = 0.1J$ and start from a) $|\psi_0\rangle = |0, 1\rangle^{\otimes L/2}$, b) $|\psi_0\rangle = |1, 2\rangle^{\otimes L/2}$.

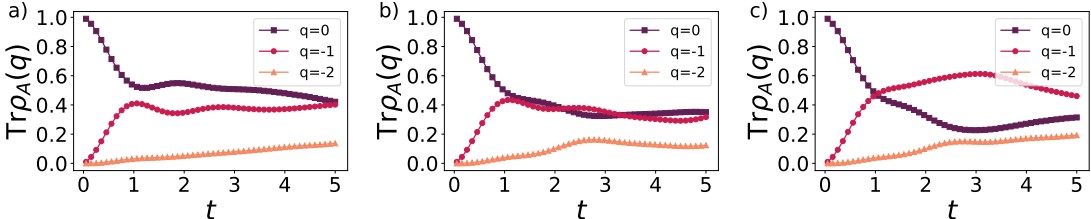

Figure 10: Population of the symmetry sectors during the evolution of the system described in Fig.6, starting from state $|\downarrow\uparrow\rangle^{\otimes N/2}$. We consider a system with $L = 12$ and we take $A = [1, 2, 3, 4, 5, 6]$ and $B = [7, 8, 9, 10, 11, 12]$. Panel a) $w = 1$, $m = 0$, $J = 1$, $\epsilon_0 = 0$; b) $w = 1$, $m = 0$, $J = 0.1$, $\epsilon_0 = 0$; c) $w = 1$ $m = 0$, $J = 1$, $\epsilon_0 = 0.5$.

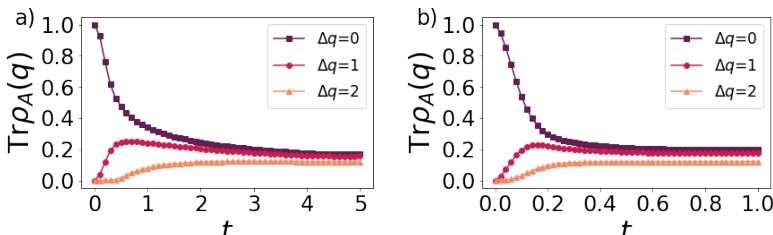

Figure 11: Population of the symmetry sectors during the evolution of the systems described in Fig.s 7a)-c). We use $\Delta q = q - \ell/2$ to label the symmetry sectors. We fix $J = 1$, $\mu = 0$. In panel a) $1D$ chain with $L = 64$, $\ell = 32$, $\gamma = 0.05$; b) $2D$ square lattice, $L = 16$, $\ell = 4$, $\gamma = 0.2$.

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
