# Peer review of "Symmetry-resolved dynamical purification in synthetic quantum matter"

_SciPost Physics, doi:SciPost Phys. 12, 106 (2022)_

## Round 2 · Referee Report · Anonymous (Referee 1) · 2022-1-3

Strengths

1-The paper introduce a procedure to analyse the quantum information content in distinct sectors of quantum systems with U(1) symmetry. 2- The theoretical calculations are done in several distinct systems, indicating the generality of their results. 3- They uses experimental results already known in the literature to corroborate their theoretical discovers. 4-There is a potential that this paper will motivate further experimental set up in the arena of quantum information.

Report

The paper certainly is in a good level and should be published

Requested changes

Revise for some trivial and minor typos.

  • validity: top
  • significance: top
  • originality: top
  • clarity: high
  • formatting: excellent
  • grammar: excellent

Author:  Vittorio Vitale  on 2022-02-14  [id 2195]

(in reply to Report 2 on 2022-01-03)

We thank the Referee for their careful reading of the manuscript, and for the feedback.
We have made an extra effort to correct a few remaining typos.

---

## Round 2 · Referee Report · Anonymous (Referee 2) · 2022-1-17

Strengths

1) Discovery of a qualitatively new event, namely that the competition between coherent and dissipative dynamics in an open many body system may lead at intermediate times to increased purity and entanglement in certain symmetry sectors although overall state becomes less pure with time. 2) Extensive set of examples demonstrating the behavior, including both numerics on a variety of realistic system and new analysis of previous experimental data.

Weaknesses

1) The restrictions on the dissipation to be compatible with the symmetry are not clarified. 2) The scaling of many of the features with the subsystem size and other parameters is not given explicitly. 3) The probability of being in a symmetry sector undergoing dynamical purification is not presented in most examples.

Report

The study of entanglement in many body systems in general, and its interplay with symmetries and conservation laws in particular, has received much attention recently, especially since the beginning of the NISQ era. To the best of my knowledge, this work presents a qualitatively new effect in this field, namely that in a many body system evolving under a combination of coherent and dissipative dynamics, while the overall purity and entanglement of a subsystem tend to decrease with time, their values in particular symmetry sectors may actually rise at intermediate times. The Authors give simple perturbative arguments for this behavior, then demonstrate it for a wide spectrum of examples, including numerics on a variety of experimentally-realizable systems, as well analysis of data from a previous experiment. The manuscript is also clearly-written. I therefore believe this work warrants publication in SciPost Phys. However, some points should be addressed first:

1) As explained by the Authors, for the effect to take place the coherent dynamics should obey the symmetry while the dissipation should break it. However, it should be noted that the dissipation should still preserve the block structure of the density matrix of a subsystem (although it should cause transitions between blocks); such a situation is known as ``weak symmetry’’, see, e.g., B. Buča and T. Prosen, A note on symmetry reductions of the Lindblad equation: Transport in constrained open spin chains, New J. Phys. 14, 073007 (2012); V. V. Albert and L. Jiang, Symmetries and conserved quantities in Lindblad master equations, Phys. Rev. A 89, 022118 (2014); and later works. The Authors should discuss the notion of weak symmetry and its role in their effect.

2) Most of the discussion concentrates on the symmetry sector which ``neighbors’’ the one which is initially populated, though it is mentioned and exemplified that similar effects may occur in other sectors. Could the Authors estimate the number of sectors in which dynamical purification may occur and its scaling with the parameters and subsystem size? It would also be useful to have explicit expressions for the scaling of the time of maximal purity, the corresponding maximal value, and the width of the peak with the various parameters and subsystem size – some sentences in Sec. 3.1 seem to indicate the Authors are somewhat reluctant to give such expressions, but I do believe that if possible, it would be useful for future readers.

3) The observability of the discussed effect depends on the probability of experimentally finding the system in a sector experiencing dynamical purification (which amounts to post-selection). However, this information is presented only for the experimental data in Sec. 6. It should be added for the analytical estimate in Sec. 3 all the numerical examples in Sec. 4.

Requested changes

1) Adding a discussion of weak symmetries and their role in the results 2) If possible, adding a discussion of the scaling of various features with the subsystem size and other parameters, especially the number of sectors experiencing dynamical purification. 3) Adding the probabilities of finding the system in a symmetry sector undergoing dynamical purification.

  • validity: high
  • significance: high
  • originality: top
  • clarity: high
  • formatting: excellent
  • grammar: excellent

Author:  Vittorio Vitale  on 2022-02-14  [id 2196]

(in reply to Report 1 on 2022-01-17)

We thank the Referee for their careful reading of the manuscript, and for their comments, that we address below point by point.

Point 1:

Indeed, as pointed out by the Referee, we are considering a system where the dissipator shall satisfy weak symmetry: for instance, in our Bose-Hubbard model, the effect of a jump operator of the form $b^\dagger +b $ would not be captured by our theory.

We have clarified this aspect in two parts of the manuscript:

a) at the beginning of Sec. 3.1, where the general conditions for the theory are presented (we have also included the references suggested by the Reviewer), where we have also emphasized that this choice reflects typical experimental conditions;

b) after Eq. 17, in a footnote we introduced.

In fact, dynamical purification can also occur even if weak symmetry is broken, as shown in our experimental example. Of course, in that case, the notion of entropy shall be handled with care, as the density matrix is not in block-diagonal form: still, symmetry resolved quantities have specific operational meaning even in that context, as we have elaborated upon in Npj Quantum Inf. 7, 152 (2021) (in particular, the offer meaningful bounds to information witnesses).

Point 2:

The Referee raises a very important point, partly connected to their third comment as well. Let us split the question into two parts - different sectors, and scaling of maximal purity.

A) In general, the principle that governs dynamical purification also applies to sectors that are not neighbors of the initially occupied one: first, dissipation scrambles information in the sector, and then, coherent dynamics partly re-order the latter. However, a clean mathematical understanding of this phenomenon is a non-trivial task.

The reason is the following: while NN sectors can be rigorously treated with second order perturbation theory, going beyond this would require at least third order, or even higher orders in 1D (since tunneling is strongly limited). This seems, in principle, technically feasible: however, empirically, we found such effects to be quite small (one example is depicted in Fig. 7), so we decided not to focus on those, as they will likely have limited experimental applicability.

B) Maximal purity: this is an extremely interesting question, that we had tried to address within out framework. Unfortunately, it turns out that the 'peak' region is something that, on its own, cannot be immediately captured by perturbation theory: it is exactly the timescale where states that belong to generic subspaces $E_2(-1), E_3(-1)$ become important, because of both (i) dynamics inside the partition and (ii) multiple dissipative events. Both of these effects are well beyond our theory, so we are unable to capture neither the position nor the maximum of the peak with precision. We are only able to determine the scaling functions (see Eq. 23). Empirically, we have observed that the position of the maximum seems relatively unaffected by noise and even interactions in both XY, Bose-Hubbard, and U(1) lattice gauge theories (Fig. 4, 5, 6), so this may suggest some generic scaling, but as it stands, this shall be understood as a numerical observation.

Point 3:

We have added the population dynamics also for some sample cases of our simulations (most models follow the same pattern) with a new appendix including 3 figures, spanning several parameter regimes. We have also commented that, in the regime of dynamical purification, the probability of being in a sector that is neighbor to the starting one is expected to grow linearly with time, and then enter a quadratic regime, similarly to the prefactors $A(t)$.

---

## Round 2 · Referee Report · Anonymous (Referee 3) · 2022-2-4

Report

This a very good paper on a timely and hot subject. The authors show that contrary to intuitions, in the presence of continuous symmetries and under ubiquitous experimental conditions, symmetry-resolved information spreading is inhibited due to the competition of coherent and incoherent dynamics: in given quantum number sectors, entropy decreases as a function of time, signalling what they call "dynamical purification". They then propose to use their random unitaries toolbox and apply to experimental date on trapped ions. This is a top class research and paper shoudl be published . I have one minor remark: to make a reference list more complete i would suggest to cite tiogehter with [90], the even newer review: 23. Monika Aidelsburger, Luca Barbiero, Alejandro Bermudez, Titas Chanda, Alexandre Dauphin, Daniel González-Cuadra, Przemysław R. Grzybowski, Simon Hands, Fred Jendrzejewski, Johannes Jünemann, Gediminas Juzeliunas, Valentin Kasper, Angelo Piga, Shi-Ju Ran, Matteo Rizzi, Gérman Sierra, Luca Tagliacozzo, Emanuele Tirrito, Torsten V. Zache, Jakub Zakrzewski, Erez Zohar, and Maciej Lewenstein, Cold atoms meet lattice gauge theory, Phil. Trans. R. Soc. A 380, 20210064 (2021), http://doi.org/10.1098/rsta.2021.0064 (2022), arXiv:2106.03063.
  • validity: -
  • significance: -
  • originality: -
  • clarity: -
  • formatting: -
  • grammar: -

Author:  Vittorio Vitale  on 2022-02-14  [id 2197]

(in reply to Report 4 on 2022-02-04)

We thank the Referee for their careful reading of the manuscript, and for the positive assessment. We have included the paper suggested to our bibliography together with the previously cited review article.

---

## Round 3 · Referee Report · Anonymous · 2022-2-22

Strengths
As in my previous report:
1) Discovery of a qualitatively new event, namely that the competition between coherent and dissipative dynamics in an open many body system may lead at intermediate times to increased purity and entanglement in certain symmetry sectors although overall state becomes less pure with time.
2) Extensive set of examples demonstrating the behavior, including both numerics on a variety of realistic system and new analysis of previous experimental data.
Weaknesses
Weaknesses identified in my previous reports were in my opinion properly addressed by the Authors.
Report
In my opinion, the revised manuscript properly addresses the comments made by all the referees on the previous version. Given my previous assessment of the importance and novelty of this work, I would now recommend its publication in SciPost Physics.
Requested changes
None

---

## Round 3 · Author Response

herewith we resubmit the paper "Symmetry-resolved dynamical purification in synthetic quantum matter" by Vittorio Vitale, Andreas Elben, Richard Kueng, Antoine Neven, Jose Carrasco, Barbara Kraus, Peter Zoller, Pasquale Calabrese, Benoit Vermersch and Marcello Dalmonte.
We thank you for the handling of our manuscript, and the Referees for the time and effort spent in reviewing it.
We include below a summary of the changes applied to the manuscript.
Sincerely yours,
Vittorio Vitale on behalf of the authors

---

## Round 3 · List of Changes

1) Clarification about weak symmetries, at the beginning of Sec. 3.1, where the general conditions for the theory are presented and after Eq. 17, in a footnote we introduced.
2) Correction of a typo in Fig. 6.
3) Correction of other minor typos in the manuscript.
4) Addition of reference "M. Aidelsburger, L. Barbiero, A. Bermudez, T. Chanda, A. Dauphin, D. González-Cuadra,P. R. Grzybowski, S. Hands, F. Jendrzejewski, J. Jünemannet al.,Cold atoms meet latticegauge theory, Philosophical Transactions of the Royal Society A380(2216), 20210064(2022)" as suggested by the Referee.
5) Appendix "Sectors Populations" and Fig.s 9-10-11, where the populations of the different symmetry sectors of the systems dynamics presented throughout the manuscript are discussed.

You are currently on this page

---

## Editorial Decision

published